

# 1 Morphodynamic model of Lower Yellow River: flux or entrainment form

# 2 for sediment mass conservation?

Chenge An[1], Andrew J. Moodie[2], Hongbo Ma[2], Xudong Fu[1], Yuanfeng Zhang[3], Kensuke Naito[4], Gary
Parker[5]
[1]Department of Hydraulic Engineering, State Key Laboratory of Hydroscience and Engineering, Tsinghua University,
Beijing, China.
[2]Department of Earth, Environmental and Planetary Sciences, Rice University, Houston, TX, USA.
[3]Yellow River Institute of Hydraulic Research, Zhengzhou, Henan, China.
[4]Department of Civil and Environmental Engineering, Hydrosystems Laboratory, University of Illinois, Urbana-Champaign,
IL, USA.
[5]Department of Civil and Environmental Engineering and Department of Geology, Hydrosystems Laboratory, University of
Illinois, Urbana-Champaign, IL, USA.
*Correspondence to*: Chenge An (anchenge08@163.com) and Xudong Fu (xdfu@tsinghua.edu.cn)
**Abstract.** Sediment mass conservation is a key factor that constrains river morphodynamic processes. In most models of river
morphodynamics, sediment mass conservation is described by the Exner equation, which may take various forms depending
on the problem in question. One of the most widely used forms of the Exner equation is the flux-based formulation, in which
the conservation of bed material is related to the streamwise gradient of the sediment transport rate. An alternate form of the
Exner equation, however, is the entrainment-based formulation, in which the conservation of bed material is related to the
difference between the entrainment rate of bed sediment into suspension and the deposition rate of suspended sediment onto
the bed. In the flux form, sediment transport is regarded to be in local equilibrium (i.e., sediment transport rate locally equals
sediment transport capacity). However, the entrainment form does not require this constraint: the sediment transport rate may
lag in space and time behind the changing flow conditions. In modeling the fine-grained Lower Yellow River, it is usual to
treat sediment conservation in terms of an entrainment (nonequilibrium) form rather than a flux (equilibrium) form, in
consideration of the condition that fine-grained sediment may be entrained at one place but deposited only at some distant
location downstream. However, the differences in prediction between the two formulations have not been comprehensively
studied to date. Here we study this problem by comparing the results of flux-based and entrainment-based morphodynamics
under conditions typical of the Lower Yellow River, but simplified for clarity of comparison. We used  sediment transport
equations specifically designed for the Lower Yellow River. We find that in a treatment of a 200 km reach using a single
characteristic bed sediment size, there is little difference between the two forms since the corresponding adaptation length is
relatively small. However, a consideration of sediment mixtures shows that the two forms give very different patterns of grain
sorting: clear kinematic waves occur in the flux form but are diffused out in the entrainment form. Both numerical simulation





and mathematical analysis show that the morphodynamic processes predicted by the entrainment form are sensitive to sediment
fall velocity.
**1. Introduction**
Models of river morphodynamics often consist of three elements (Parker, 2004): (1) a treatment of flow hydraulics;
(2) a formulation relating some aspect of sediment transport to flow hydraulics; and (3) a description of sediment conservation.
In the case of unidirectional river flow, the Exner equation of sediment conservation has usually been described in terms of a
flux-based form in which temporal bed elevation change is related to the streamwise gradient of the sediment transport rate
$\partial q_s / \partial x$, where $q_s$ is the total volumetric sediment transport rate per unit width and $x$ is the streamwise coordinate (Exner, 1920;
Parker et al., 2004). This formulation is also referred to as the equilibrium formulation, since it considers sediment transport
to be at local equilibrium, i.e. $q_s$ equals its sediment transport capacity $q_{se}$, regardless of the variation of flow conditions. Under
this assumption, sediment transport relations developed under equilibrium flow conditions (e.g., Meyer-Peter and Müller, 1948;
Engelund and Hansen, 1967; Brownlie, 1981) can be incorporated directly in such a formulation to calculate $q_s$, which is
related to one or more flow parameters such as bed shear stress.
An alternate formulation, however, is available in terms of an entrainment-based form of the Exner equation, in which
bed elevation variation is related to the difference between the entrainment rate of bed sediment into suspension and the
deposition rate of suspended sediment on the bed (Parker, 2004). The basic idea of the entrainment formulation can be traced
back to Einstein (1937)'s pioneering work of bedload transport, and has been developed since then by numerous researchers
so as to treat either bedload or suspended load (Tsujimoto, 1978; Armanini and Di Silvio, 1988; Parker et al., 2000; Wu and
Wang, 2008; Guan et al., 2015). Such a formulation differs from the flux formulation in that it is the rate of entrainment of bed
sediment, rather than the sediment transport rate itself, that is related to flow hydraulics. The difference between the local
entrainment rate from the bed and the local deposition rate onto the bed determines the rate of bed aggradation/degradation,
and concomitantly the rate of loss/gain of sediment in motion in the water column. Therefore, the sediment transport rate is no
longer assumed to be in an equilibrium transport state, but may exhibit lags in space and time after changing flow conditions.
The entrainment formulation is also referred to as the nonequilibrium formulation (Armanini and Di Silvio, 1988; Wu and
Wang, 2008; Zhang et al., 2013).
To describe the lag effects between sediment transport and flow conditions, the concept of an adaptation length/time
is widely applied. This length/time characterizes the distance/time for sediment transport to reach its equilibrium state (i.e.,
transport capacity). Using the concept of the adaptation length, the Exner equation can be recast into a first-order "reaction"
equation, in which the deformation term is related to the difference between the actual and equilibrium sediment transport
rates, as mediated by an adaptation length (which can also be recast as an adaptation time). (Armanini and Di Silvio, 1988;
Wu and Wang, 2008; Minh Duc and Rodi, 2008; El kadi Abderrezzak and Paquier, 2009). The adaptation length is thus an
important parameter for bed evolution under nonequilibrium sediment transport conditions, and various estimates have been



proposed. For suspended load, the adaptation length is typically calculated as a function of flow depth, flow velocity and
sediment fall velocity (Armanini and Di Silvio, 1988; Wu et al., 2004; Wu and Wang, 2008; Dorrell and Hogg, 2012; Zhang
et al., 2013). The adaptation length of bedload, on the other hand, has been related to a wide range of parameters, including
the sediment grain size (Armanini and Di Silvio, 1988), the saltation step length (Phillips and Sutherland, 1989), the dimensions
of particle diffusivity (Bohorquez and Ancey, 2016), the length of dunes (Wu et al., 2004), and the magnitude of a scour hole
formed downstream of an inerodible reach (Bell and Sutherland, 1983). For simplicity, the adaptation length can also be
specified as a calibration parameter in river morphodynamic models (El kadi Abderrezzak and Paquier, 2009; Zhang and Duan,
2011). Nonetheless, no comprehensive definition of adaptation length exists.

In this paper we apply the two forms of the Exner equation mentioned above to the Lower Yellow River (LYR) in

China. The LYR describes the river section between Tiexie and the river mouth, and has a total length of about 800 km. Figure
1(a) shows a sketch of the LYR along with 6 major gauging stations and the Xiaolangdi Dam, which is 26 km upstream of
Tiexie. The LYR has an exceptionally high sediment concentration (Ma et al., 2017), historically exporting more than 1 Gt of
sediment per year with only 49 billion tons of water, leading to a sediment concentration an order of magnitude higher than
most other large lowland rivers worldwide (Milliman and Meade, 1983; Ma et al., 2017; Naito et al., accepted subject to
revision). More recently, however, since the operation of Xiaolangdi Dam in 1999 the LYR has seen a substantial reduction
in its sediment load (Fig. 1(b)) because most of the sediment load is derived from the river reach upstream of the reservoir,
especially from the Loess Plateau (Wang et al., 2016; Naito et al., accepted subject to revision). Finally, the bed material of
the LYR is very fine, ranging as low as 15 μm. This is much finer than the conventional cutoff of washload (62.5 μm) employed
for sediment transport in most sand-bed rivers (National Research Council, 2007; Ma et al., 2017).



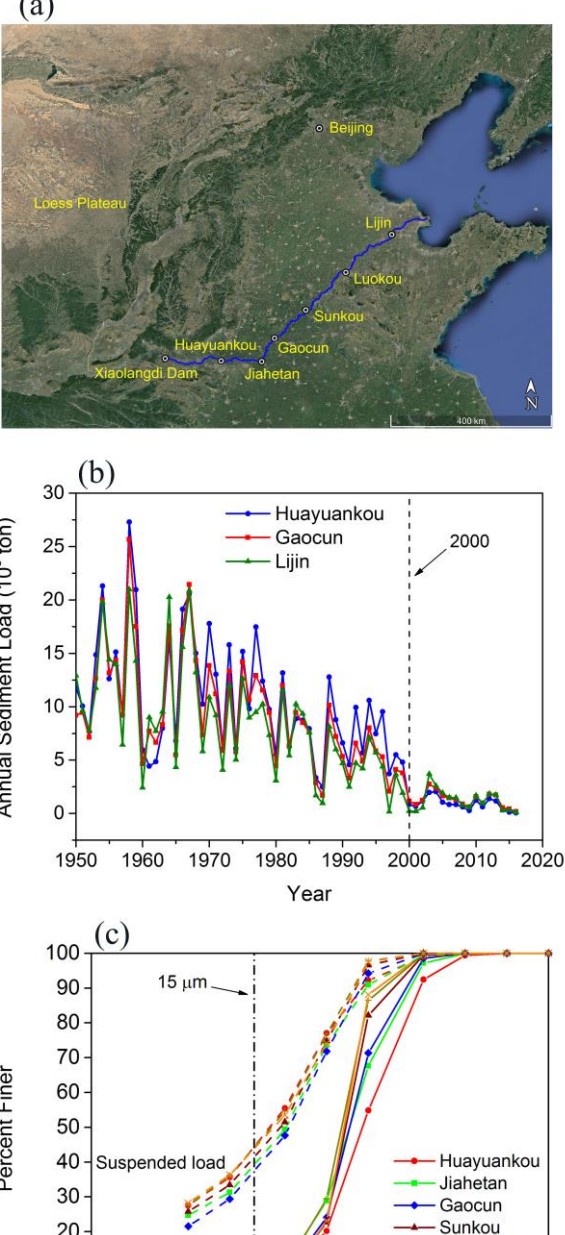

**Figure 1.** (a) Sketch of Lower Yellow River, showing 6 major gauging stations and the Xiaolangdi Dam; (b) Annual sediment
load of LYR measured at 3 gauging stations since 1950; (c) Grain size distributions of both bed material and suspended load
measured at 6 gauging stations of the LYR.






When modeling the high-concentration and fine-grained LYR, it is common to treat sediment conservation in terms

of an entrainment-based rather than a flux-based formulation. This is because many Chinese researchers view the entrainment
formulation as more physically based, and thus more likely to be capable of describing the behavior of fine-grained sediment,
which when entrained at one place may be deposited at some distant location downstream (Zhang et al., 2001; Ni et al., 2004;
Cao et al., 2006; He et al., 2012; Guo et al., 2008). However, the entrainment formulation is more computationally expensive
and more complex to implement. In so far as the differences in prediction between the two formulations do not appear to have
been studied in a systematic way, here we pose our central questions. Is the entrainment formulation really necessary when
modeling the LYR? Or more specifically, under what circumstances should a numerical modeler be impelled to implement the
entrainment formulation instead of the flux formulation for river morphodynamic modeling?

Here we study this problem by comparing the results of flux-based and entrainment-based morphodynamics under

conditions typical of the LYR. The organization of this paper is as follows. The numerical model is described in Section 2. In
Section 3, the model is implemented to predict the morphodynamics of the LYR. We find that the two forms of the Exner
equation give similar predictions in the case of uniform sediment, but show different sorting patterns in the case of sediment
mixtures. In Section 4, we conduct a mathematical analysis to explain the results in Section 3, and more specifically we quantify
the effects of varied sediment fall velocity in the simulations. Finally, we summarize our conclusions in Section 5.
**2. Model formulation**

In this paper, we present a one-dimensional morphodynamic model for the Lower Yellow River. The fully unsteady

Saint Venant Equations are implemented for the hydraulic calculation. As the main topic of this paper is to compare the flux
form and entrainment form of Exner equation, both the formulations of the Exner equation for sediment conservation are
implemented. For each formulation, we consider both the cases of uniform sediment (bed material characterized by a single
grain size) and sediment mixtures. Since the sediment is very fine in the LYR, the component of the load that is bedload is
likely negligible (e.g. Ma et al., 2017), so that we consider only the transport of suspended load. Considering the fact that most
well accepted sediment transport relations (e.g., the Engelund and Hansen. 1967 relation) underpredict the sediment transport
rate of the LYR by an order of magnitude or more (Ma et al., 2017), in our model we implement two recently developed
generalized versions of the Engelund-Hansen relation which are based on data from the LYR. These are the version of Ma et
al. (2017) for uniform sediment, and the version of Naito et al. (accepted subject to revision) for sediment mixtures. In cases
considering sediment mixtures, we also implement the method of Viparelli et al. (2010) to store and access bed stratigraphy
as the bed aggrades and degrades.

Some additional simplifications to the model facilitate comparison across the Exner formulation model runs. The

channel is simplified to be a constant-width rectangular channel, and bank (sidewall) effects and floodplain interactions are
not considered. The channel bed is assumed to be an infinitely deep supplier of erodible sediment with no exposed bedrock.



Finally, water and sediment are fed into the upstream boundary at a specified rate, and at the downstream end of the channel
we specify a fixed bed elevation along with a normal flow depth. These restrictions could be easily relaxed so as to incorporate
site-specific complexities of the Yellow River.

## 2.1 Flow hydraulics

Flow hydraulics in a rectangular channel is described by the following 1D Saint Venant equations, which consider
the fluid mass and momentum conservation,

$$\frac{1}{I_f}\frac{\partial h}{\partial t} + \frac{\partial q_w}{\partial x} = 0 \tag{1}$$

$$\frac{1}{I_f}\frac{\partial q_w}{\partial t} + \frac{\partial}{\partial x}\left(\frac{q_w^{\,2}}{h} + \frac{1}{2}gh^2\right) = ghS - C_f u^2 \tag{2}$$

$$C_f = C_z^{-2} \tag{3}$$

where $t$ is time, $h$ is water depth, $q_w$ is flow discharge per unit width, $g$ is gravitational acceleration, $S$ is bed slope, $u$ is depth-
averaged flow velocity, $C_f$ is dimensionless bed resistance coefficient, and $C_z$ is the dimensionless Chezy resistance coefficient.
In our model, the fully unsteady 1D Saint Venant equations are solved using a Godunov type scheme with the HLL (Harten-
Lax-van Leer) approximate Riemann solver (Harten et al., 1983; Toro, 2001), which can effectively capture discontinuities in
unsteady and nonuniform open channel flows.
In this paper, the full flood hydrograph of the LYR is replaced by a flood intermittency factor $I_f$ (Paola et al., 1992;
Parker, 2004). According to this definition, the river is assumed to be at low flow and not transporting significant amounts of
sediment for time fraction $1 - I_f$; and is in flood at constant discharge and active morphodynamically for time fraction $I_f$. In the
long term, the relation between the flood time scale $t_f$ and the actual time scale $t$ is $t_f = I_f t$. For all the governing equations in
this paper, the flood time scale is implemented by introducing $I_f$ into each time derivative.

## 2.2 Flux form of the Exner equation

When dealing with uniform sediment, the flux form of the Enxer equation can be written as,

$$\frac{1}{I_f}\left(1-\lambda_p\right)\frac{\partial z_b}{\partial t} = -\frac{\partial q_s}{\partial x} \tag{4}$$



where $\lambda_p$ is the porosity of the bed deposit, taken to be 0.4 in this paper and $z_b$ is bed elevation. Sediment transport is regarded
to be in a quasi-equilibrium state, so that the sediment transport rate per unit width $q_s$ equals the equilibrium (capacity) sediment
transport rate per unit width $q_{se}$.
When considering sediment mixtures, an active layer formulation (Hirano, 1971; Parker, 2004) is incorporated in the
flux-based Exner equation, so that the evolution of both bed elevation and surface grain size distribution can be considered. In
this formulation, the river bed is divided into a well-mixed upper active layer and a lower substrate with vertical stratigraphic
variations. The upper active layer therefore represents the volume of sediment that interacts directly with suspended load
transport, and also exchanges with the substrate as the bed aggrades and degrades. Discretizing the grain size distribution into
$n$ ranges, the mass conservation relation for each grain size range can be written as,
$$\frac{1}{I_f}\left(1-\lambda_p\right)\left[f_{Ii}\frac{\partial}{\partial t}\left(z_b-L_a\right)+\frac{\partial}{\partial t}\left(F_i L_a\right)\right]=-\frac{\partial q_{si}}{\partial x} \tag{5}$$

where $q_{si}$ is volumetric sediment transport rate per unit width of the $i$-th grain size range ( taken to be equal to its equilibrium
value $q_{sei}$ in the flux formulation), $F_i$ is the volumetric fraction of surface material in the $i$-th grain size range; $f_{Ii}$ is volumetric
fraction of material in the $i$-th grain size range exchanged across the surface-substrate interface as the bed aggrades or degrades,
and $L_a$ is the thickness of active layer. For bedform-dominated sand-bed rivers, $L_a$ is often related to the height of dunes so that
the vertical sorting processes due to bedform migration can be considered (Blom et al., 2003). In this paper, a constant value
of $L_a$ is implemented in the simulation.
Summing Eq. (5) over all grain size ranges, one can find that the governing equation for bed elevation in case of
sediment mixtures is the same as Eq. (4) upon replacing $q_s$ with $q_{sT}=\Sigma q_{si}$, where $q_{sT}$ denotes the total sediment transport rate
per unit width summer over all size ranges. Reducing Eq. (5) with Eq. (4) we get,
$$\frac{1}{I_f}\left(1-\lambda_p\right)\left[L_a\frac{\partial F_i}{\partial t}+\left(F_i-f_{Ii}\right)\frac{\partial L_a}{\partial t}\right]=f_{Ii}\frac{\partial q_{sT}}{\partial x}-\frac{\partial q_{si}}{\partial x} \tag{6}$$

Therefore, in the flux formulation Eqs. (4) and (6) are implemented as governing equations for sediment mixtures,
with Eq. (4) describing the evolution of bed elevation and Eq. (6) describing the evolution of surface grain size distribution.
The exchange fractions $f_{Ii}$ between the active layer and the substrate are calculated using the following closure relation,
$$f_{Ii}=\begin{cases}f_i\mid_{z_b-L_a} & \dfrac{\partial z_b}{\partial t}<0 \\[2ex] \alpha F_i+\left(1-\alpha\right)p_{si} & \dfrac{\partial z_b}{\partial t}>0\end{cases} \tag{7}$$



That is, the substrate is transferred into the active layer during degradation, and a mixture of suspended load and active layer
material is transferred into substrate during aggradation. In Eq. (7), $f_i|_{zb-La}$ is the volumetric fraction of substrate material just
beneath the interface, $p_{si} = q_{si}/q_{sT}$ is the fraction of bed material load in the $i$-th grain size range, and $\alpha$ is a specified parameter
between 0 and 1 The formulation is adapted from Hoey and Ferguson (1994) and Toro-Escobar et al. (1996), who originally
used it for bedload. In this paper, a value of 0.5 is specified for $\alpha$.
The method of Viparelli et al. (2010) is applied in our model to store substrate stratigraphy and provide information
for $f_i|_{zb-La}$ (i.e., the topmost sublayer in Viparelli et al., 2010). The reader can refer to the original reference of Viparelli et al.
(2010) for more details, or refer to An et al. (2017) for a concise description as how to implement this method in a
morphodynamic model.

**2.3 Entrainment form of the Exner equation**

The entrainment-based Exner equation for uniform sediment is,

$$\frac{1}{I_f}\left(1 - \lambda_p\right)\frac{\partial z_b}{\partial t} = -v_s\left(E - r_0 C\right) \tag{8}$$

In Eq. (8), $v_s$ is the fall velocity of sediment particles; $E$ is the dimensionless entrainment rate of sediment normalized by
sediment fall velocity; $C$ is the depth-flux-averaged volume sediment concentration; and $r_o = c_b/C$ is the recovery coefficient
of suspended load which denotes the ratio between the near-bed sediment concentration $c_b$ and the flux-averaged sediment
concentration $C$. By definition, $r_0$ is related to the concentration profile of suspended load, and is expected to be no less than
unity in cases appropriate for a depth-averaged shallow-water treatment of flow and morphodynamics.
For sediment fall velocity $v_s$, we compare two widely used relations: the relation of Dietrich (1982), and the relation
of Ferguson and Church (2004). Results show that these two relations give almost the same fall velocity for bed material load
of the LYR, whose grain sizes typically fall in the rage of 15 μm to 500 μm. Therefore, only the relation of Dietrich (1982) is
implemented in our simulations in this paper. Readers can refer to Appendix A for more details.
Since sediment transport is not necessarily in its equilibrium state in the entrainment formulation, we relate the
sediment entrainment rate, rather than the sediment transport rate, to the equilibrium sediment transport rate. Thus

$$E = r_0 \frac{q_{se}}{q_w} \tag{9}$$

For the depth-flux-averaged sediment concentration $C$, another equation is implemented describing the conservation of
suspended sediment in the water column,





$$\frac{1}{I_f}\frac{\partial(hC)}{\partial t}+\frac{\partial(huC)}{\partial x}=v_s\left(E-r_0C\right)$$ (10)
Note that sediment transport is at equilibrium when $E = r_oC$. The sediment transport rate per unit width $q_s$ obeys a continuity
relation,
$$q_s = huC$$ (11)

195   The entrainment-form Exner equation for sediment mixtures also uses the active layer formulation described in

Section 2.2. Mass conservation of each grain size range can be written as,
$$\frac{1}{I_f}\left(1-\lambda_p\right)\left[f_{Ii}\frac{\partial}{\partial t}\left(z_b-L_a\right)+\frac{\partial}{\partial t}\left(F_iL_a\right)\right]=-v_{si}\left(E_i-r_{0i}C_i\right)$$ (12)
$$E_i = r_{0i}\frac{q_{sei}}{q_w}$$ (13)
where the subscript $i$ denotes the $i$-th size range of sediment grain size.

200   Summing Eq. (12) over all grain size ranges, we get the governing equation for bed elevation,

$$\frac{1}{I_f}\left(1-\lambda_p\right)\frac{\partial z_b}{\partial t}=-\sum_{j=1}^{n}v_{sj}\left(E_j-r_{0j}C_j\right)$$ (14)
Reducing Eq. (12) with Eq. (14) we get the governing equation for surface fraction $F_i$,
$$\frac{1}{I_f}\left(1-\lambda_p\right)\left[L_a\frac{\partial F_i}{\partial t}+\left(F_i-f_{Ii}\right)\frac{\partial L_a}{\partial t}\right]=f_{Ii}\sum_{j=1}^{n}v_{sj}\left(E_j-r_{0j}C_j\right)-v_{si}\left(E_i-r_{0i}C_i\right)$$ (15)

204   The governing equation for the sediment concentration of each grain size $C_i$ can be written as,

$$\frac{1}{I_f}\frac{\partial(hC_i)}{\partial t}+\frac{\partial(huC_i)}{\partial x}=v_{si}\left(E_i-r_0C_i\right)$$ (16)
and the sediment transport rate per unit width for the $i$-th size range $q_{si}$ obeys the following continuity relation,
$$q_{si} = huC_i$$ (17)

208   In the entrainment formulation, the closure relation for $f_{Ii}$ is the same as that used in the flux formulation (i.e., Eq.

(7)), and the substrate stratigraphy is also stored and accessed using the method of Viparelli et al. (2010).




**2.4 Sediment transport relation**

**2.4.1 Uniform sediment**

To close the Exner equations described in Sections 2.2 and 2.3, equations for equilibrium sediment transport rate $q_{se}$ ($q_{sei}$) are still needed. For the simulations using uniform sediment, we implement the generalized Engelund-Hansen relation proposed by Ma et al. (2017). This equation is based on the data from LYR and can be written in the following dimensionless form,

$$q_s^* = \frac{\alpha_s}{C_f}\left(\tau^*\right)^{n_s}$$

(18)

where $q_s^*$ is dimensionless sediment transport rate per unit width (i.e., the Einstein number), and $\tau^*$ is dimensionless shear stress (i.e., the Shields number). They are defined as,

$$q_s^* = \frac{q_{se}}{\sqrt{RgD}D}$$

(19)

$$\tau^* = \frac{\tau_b}{\rho RgD}$$

(20)

$$\tau_b = \rho C_f u^2$$

(21)

where $D$ is characteristic grain size of the bed sediment (here approximated as uniform); $\tau_b$ is bed shear stress; and $R$ is submerged specific gravity of sediment, defined as $(\rho_s - \rho)/\rho$, in which $\rho_s$ is density of sediment, and $\rho$ is density of water. The sediment submerged specific gravity $R$ is specified as 1.65 in this paper, which is an appropriate estimate for natural rivers, and corresponds to quartz.

In the relation of Ma et al. (2017), the dimensionless coefficient $\alpha_s = 0.9$ and the dimensionless exponent $n_s = 1.68$. These values are quite different from the original relation of Engelund and Hansen (1967), in which $\alpha_s = 0.05$ and $n_s = 2.5$. Ma et al. (2017) demonstrated that such differences imply that the riverbed of the LYR is dominated by low-amplitude bedform features (dunes) approaching upper-regime plane bed. According to this finding, form drag is then neglected in our modeling, and all of the bed shear stress is used for sediment transport.

**2.4.2 Sediment mixtures**

We implement the relation of Naito et al. (accepted subject to revision) to calculate the equilibrium sediment transport rate of size mixtures. Using field data from the LYR, Naito et al. (accepted subject to revision) extended the Engelund and





Hansen (1967) relation to a surface-based grain-size specific form, in which the suspended load transport rate of the *i*-th size
range is tied to the availability of this size range on bed surface:
$$q_{sei} = \frac{N_i^* F_i u_*^3}{RgC_f}$$                   (22)
where $N_i^*$ is the dimensionless sediment transport rate in the *i*-th size range, and $u_*$ is shear velocity calculated from the bed
shear stress $\tau_b$:
$$u_* = \sqrt{\frac{\tau_b}{\rho}}$$                   (23)

The transport relation itself takes the form,

$$N_i^* = A_i \left( \tau_g^* \frac{D_{sg}}{D_i} \right)^{B_i}$$                   (24)
in which $D_i$ is the characteristic grain size for sediment in the *i*-th size range, $D_{sg}$ is the geometric mean grain size in the active
layer, and $\tau_g^*$ is the dimensionless bed shear stress associated with $D_{sg}$. The parameters $\tau_g^*$, coefficient $A_i$, and exponent $B_i$ are
calculated as,
$$\tau_g^* = \frac{\tau_b}{\rho RgD_{sg}}$$                   (25)
$$A_i = 0.46 \left( \frac{D_i}{D_{sg}} \right)^{-0.84}$$                   (26)
$$B_i = 0.35 \left( \frac{D_i}{D_{sg}} \right)^{-1.16}$$                   (27)

The forms of Eqs. (26) and (27) indicate that the hiding effects between coarse and fine sediment play a role in this

sediment transport relation.



## 3. Numerical modeling of the LYR using the two forms of Exner equation


In this section, we conduct numerical simulations using both the flux form and the entrainment form of the Exner
equation, with the aim to study under what circumstances the two forms give different predictions. Numerical simulations are
conducted in the setting of the LYR. We specify a 200 km long channel reach for our simulations, along with a constant
channel width of 300 m and an initial longitudinal slope of 0.0001. Bed porosity $\lambda_p$ is specified as 0.4. Based on field
measurements of the LYR available to us, we implemented a dimensionless Chezy resistance coefficient $C_z$ of 30, which
corresponds to a dimensionless bed resistance coefficient $C_f$ of 0.0011. For the entrainment form of Exner equation, we specify
the ratio of near bed sediment concentration to flux-averaged sediment concentration $r_0$ ($r_{0i}$) = 1. Such a value of $r_0$ ($r_{0i}$)
corresponds to a vertically uniform profile of sediment concentration, and will thus give a maximum difference between the
prediction of entrainment form and the prediction of the flux form. More discussion about the effects of $r_0$ will be presented
in Section 4.3.
A constant flow discharge of 2000 m³/s (corresponding to a flow discharge per unit width $q_w$ of 6.67 m²/s) is
introduced at the inlet of the channel with the flood intermittency factor $I_f$ estimated as 0.14 (Naito et al., accepted subject to
revision). The downstream end is specified far from the river mouth to neglect the effects of backwater. Therefore, the bed
elevation is held constant and the water depth is specified as the normal flow depth at the downstream end of the calculational
domain. The above flow discharge per unit width $q_w$ combined with the bed slope $S$ as well as the bed resistance coefficient $C_f$
leads to a normal flow depth of 3.69 m. In our simulation, we use the height of bedforms in the LYR to determine the thickness
of the active layer (Blom et al., 2003). According to the field survey of Ma et al. (2017), the characteristic height of bedforms
in the LYR is about 20% of the normal flow depth, which can fall in the range suggested by the data analysis of Bradley and
Venditti (2017). This eventually leads to an estimate of active layer thickness of $L_a = 0.738$ m.
Two cases are considered here. In the first case, the sediment grain size distribution of LYR is simplified to a uniform
grain size of 65 μm. This is based on the measured grain size distribution of bed material at the Lijin gauging station, which
has a median grain size of $D_{50} = 66.6$ μm and a geometric mean grain size of $D_g = 65.5$ μm, as shown in Fig. 1(c). In the second
case, we consider the effects of sediment mixtures. The grain size distribution of the initial bed is based on the bed material at
the Lijin gauging station as shown in Fig. 1(c), but we renormalize the measured grain size distribution with a cutoff for
washload at 15 μm as suggested by Ma et al. (2017). The renormalized grain size distribution for the initial bed as implemented
in the case of sediment mixtures is shown in Fig. 2. In both the two cases, simulations start with an equilibrium state where
sediment supply rate = sediment transport rate = equilibrium sediment transport rate, so that the initial state of the channel is
in equilibrium. Then we cut the sediment supply rate (of each size range) to only 10% of the equilibrium sediment transport
rate and keep this sediment supply rate. This is to mimic the reduction of sediment load in the LYR in recent years, as shown
in Fig. 1(b). The grain size distribution of sediment supply in the case of sediment mixtures is shown in Fig. 2.
The 200 km channel reach is discretized into 401 cells, with cell size $\Delta x$ of 500 m. In the case of uniform sediment,
we specify a time step for morphologic calculation $\Delta t_m = 10^{-4}$ year and a time step for hydraulic calculation $\Delta t_h = 10^{-6}$ year. In





the case of sediment mixtures, we specify a time step for morphologic calculation $\Delta t_m = 10^{-5}$ year, and a time step for hydraulic
calculation $\Delta t_h = 10^{-6}$ year. Computational conditions are briefly summarized in Table 1. The computational conditions we
implement are much simpler than the rather complicated conditions of the actual LYR. But it should be noted that the aim of
this paper is not to reproduce specific aspects of the morphodynamic processes of LYR, but to compare the flux form and
entrainment form of Exner equation in the context of conditions typical of LYR.

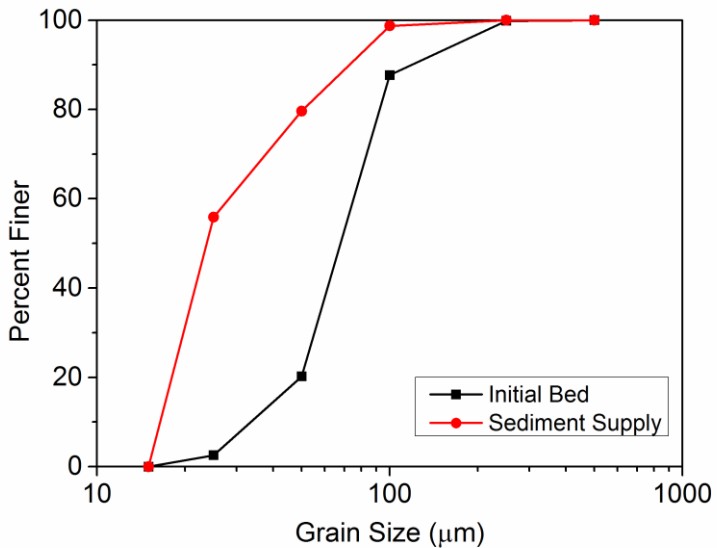

**Figure 2.** Grain size distributions of both the initial bed and the sediment supply in the case of sediment mixtures. The grain
size distribution of the initial bed is renormalized based on the field data at the Lijin gauging station. The grain size distribution
of the sediment supply equals to the grain size distribution of bed material load at equilibrium. Washload sizes have been
removed from both distributions.
**Table 1.** Summary of computational conditions for numerical modeling of the LYR.

| Parameter | Value |
|---|---|
| Channel length $L$ | 200 km |
| Channel width $B$ | 300 m |
| Initial slope $S_I$ | 0.0001 |
| Dimensionless Chezy resistance coefficient $C_z$ | 30 |
| Flow discharge per unit width $q_w$ | 6.67 m²/s |
| Flood intermittency factor $I_f$ | 0.14 |
| ratio of near bed concentration to average concentration $r_0$ ($r_{0i}$) | 1 |
| Characteristic grain size in the case of uniform sediment | 65 μm |
| Submerged specific gravity of sediment $R$ | 1.65 |
| Porosity of bed deposits $\lambda_p$ | 0.4 |




### 3.1 Case of uniform sediment

In this case, we implement a uniform grain size of 65 μm for both the bed material and sediment supply. Such a grain size is nearly equal to the observed median grain size (or geometric mean grain size) of bed material at Lijin gauging station. The relation of Ma et al. (2017) is implemented to calculate the transport rate of bed material suspended load. This relation provides an equilibrium sediment transport rate per unit width $q_{se}$ of 0.0136 m$^2$/s under the given flow discharge, bed slope and sediment grain size. With a flood intermittency factor $I_f$ of 0.14, this further gives a mean annual bed material load of 47.8 Mt/a. Adding in washload according to the estimate of Naito et al. (accepted subject to revision), total mean annual load is 86.9 Mt/a, a value that is of the same order of magnitude as averages over the period 2000-2016 (89-126 Mt/a depending on site), i.e. since the operation of Xiaolangdi Dam in 1999 (Fig. 1(b)). The sediment supply rate $q_{sf}$ we specify at the upstream end of the channel is only 10% of the equilibrium sediment transport rate (i.e. sediment supply rate is cut by 90% from the equilibrium state), such that $q_{sf} = 0.00136$ m$^2$/s.

Figure 3 shows the modeling results using the flux form of the Exner equation. As we can see in the figure, the bed degrades and the sediment load decreases in response to the cutoff of sediment supply. Such adjustments start from the upstream end of the channel and gradually migrate downstream. Figure 4 shows the modeling results using the entrainment form of Exner equation. A comparison between Fig. 4 and Fig. 3 shows that the entrainment form and the flux form give very similar predictions in this case. The entrainment form provides a somewhat slower degradation and a more diffusive sediment load reduction. Such more diffusive predictions of sediment load variation can be ascribed to the concept of nonequilibrium transport that is embedded in the entrainment form. This issue will be studied analytically in Section 4. Here we present the results of only 0.2 year after the cutoff of sediment supply, since the differences between the predictions of the two forms tend to be the most evident shortly after the disruption but gradually diminish as the river approaches the new equilibrium (El kadi Abderrezzak and Paquier, 2009). Modeling results over a longer time scale will be discussed in Section 4.3.





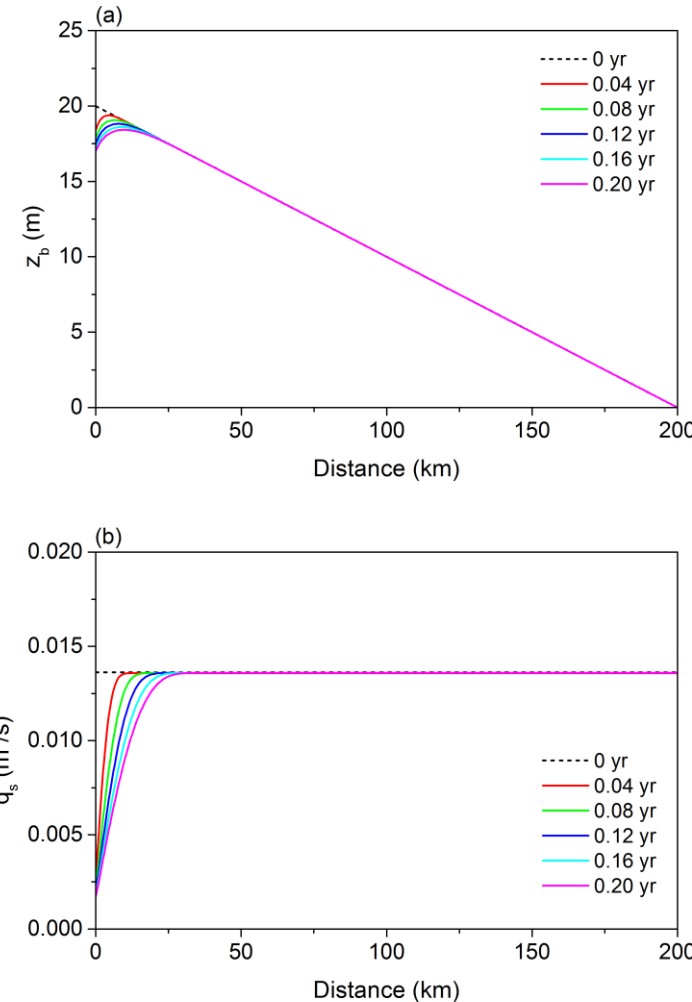

**Figure 3.** 0.2 year results for the case of uniform sediment using the flux form of Exner equation: time variation of (a) bed
elevation $z_b$ and (b) sediment load per unit width $q_s$ of the LYR in response to the cutoff of sediment supply.



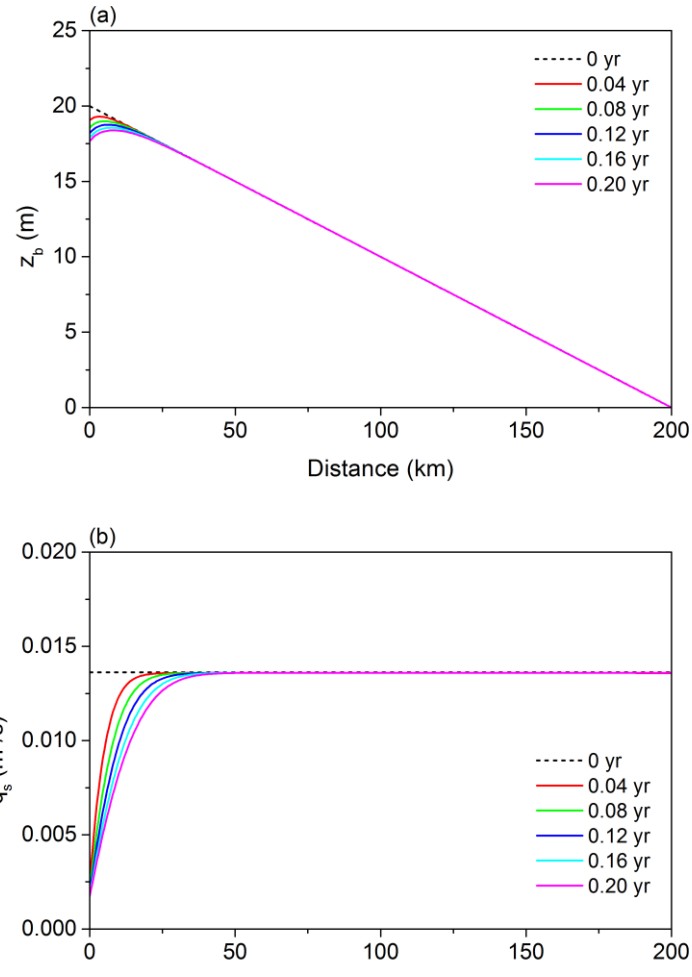

**Figure 4.** 0.2 year results for the case of uniform sediment using the entrainment form of Exner equation: time variation of (a)
bed elevation $z_b$ and (b) sediment load per unit width $q_s$ of the LYR in response to the cutoff of sediment supply.

To further quantify the differences between the predictions of the two forms, we propose the following normalized

parameter,
$$\delta(y) = \left| \frac{y_E - y_F}{y_F} \right| \times 100\% \tag{28}$$
where $y$ denotes an arbitrary variable calculated by the morphodynamic model, and subscripts $F$ and $E$ denote results using the
flux form and the entrainment form respectively. Therefore, $\delta(y)$ denotes the difference between the prediction the two forms
$y_F$ and $y_E$ normalized by the prediction of the flux form $y_F$.





Table 2 gives a summary of the maximum values of $\delta$ along the channel at different times in the case of uniform

sediment. The values of $\delta$ for both $z_b$ and $q_s$ are presented. As we can see from the table, the maximum value of $\delta(z_b)$ along the

calculational domain keeps within 4% in the first 0.2 year after the cutoff of sediment supply. This indicates that the flux form

and the entrainment form can indeed give almost the same prediction in terms of bed elevation in this case. But in the case of

the sediment load per unit width $q_s$, the maximum value of $\delta(q_s)$ can be as high as 20%, indicating that even though the two

forms give qualitatively similar patterns of evolution in terms of sediment load as shown in Figs. 3 and 4, the quantitative

difference can be clearly evident due to the more diffusive nature of the predictions of the entrainment form. The value of $\delta(q_s)$

is largest at the beginning of the simulation, and then gradually reduces with time.

**Table 2.** Quantification of the difference between predictions of the flux form and the entrainment form in the case of uniform

sediment. The maximum values of $\delta(z_b)$ and $\delta(q_s)$ in the calculational domain are presented every 0.04 year.

| | | 0.04 yr | 0.08 yr | 0.12 yr | 0.16 yr | 0.20 yr |
|---|---|---|---|---|---|---|
| original $v_s$ | $\delta(z_b)$ | 3.66 % | 3.91 % | 3.93 % | 3.88 % | 3.81 % |
| | $\delta(q_s)$ | 20.48 % | 15.11 % | 12.31 % | 10.48 % | 9.17 % |
| $v_s$ multiplied by 0.05 | $\delta(z_b)$ | 8.23 % | 10.94 % | 12.66 % | 13.92 % | 14.91 % |
| | $\delta(q_s)$ | 74.83 % | 68.14 % | 63.04 % | 58.89 % | 55.41 % |

The above results show that the flux form and the entrainment form can provide similar predictions of LYR when the

bed sediment grain size distribution is simplified to a uniform value of 65 μm. To understand under what conditions the two

forms will lead to more different results, we conduct an idealized run using the entrainment form in which the sediment fall

velocity $v_s$ is arbitrarily multiplied by a factor of 0.05. That is to say, we keep the sediment grain size at 65 μm in the

computation of the Shields number, but let the sediment fall velocity in Eqs. (8) and (10) equal only 1/20 of the value calculated

by the relation of Dietrich (1982) from this grain size. With a much smaller, and indeed intentionally unrealistic sediment fall

velocity, the entrainment form predicts very different results as shown in Fig. 5. The adjustments of the sediment load become

even more diffusive in space: it almost takes the entire 200 km reach for the sediment load to adjust from the upstream

disruption to the equilibrium transport rate. Meanwhile, there is barely any bed degradation at the upstream end after 0.2 year,

in correspondence with the fact that the spatial gradient of $q_s$ becomes quite small. In Table 2 we also exhibit the $\delta$ values for

this idealized run. It is no surprise that both $\delta(z_b)$ and $\delta(q_s)$ are high, as the entrainment form and flux form predict very different

patterns with such an arbitrarily reduced sediment fall velocity.



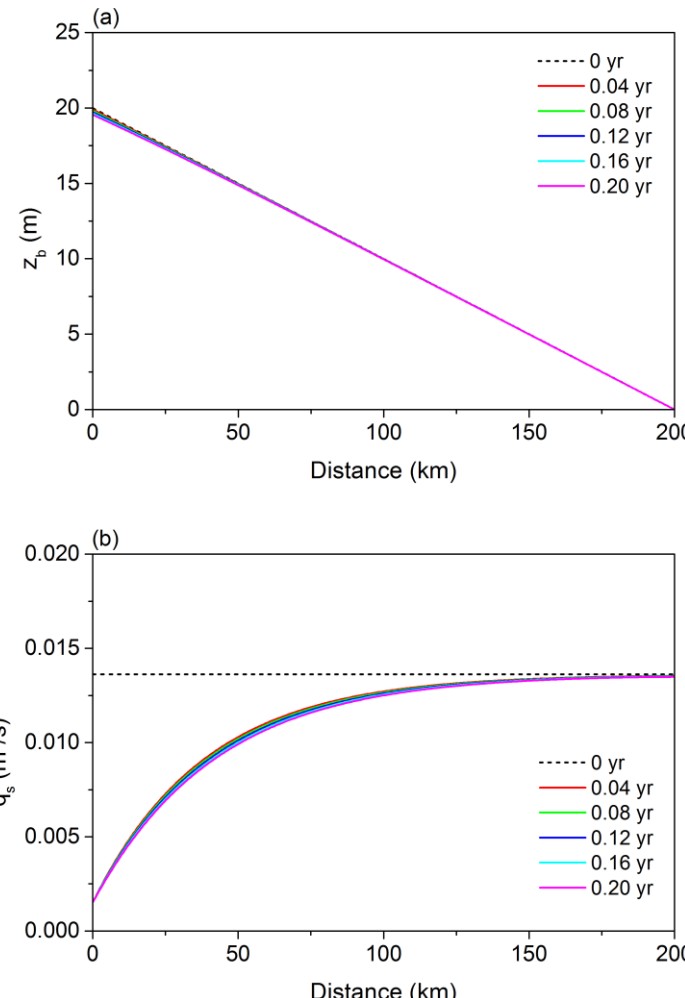

**Figure 5.** 0.2 year results for the case of uniform sediment using the entrainment form of Exner equation: time variation of (a)
bed elevation $z_b$ and (b) sediment load per unit width $q_s$ of the LYR in response to the cutoff of sediment supply. Sediment fall
velocity $v_s$ is arbitrarily multiplied by a factor of 0.05 while holding bed grain size constant in this run.
**3.2 Case of sediment mixtures**
In this section we consider the morphodynamics of sediment mixtures rather than the case of a uniform bed grain size
implemented in section 3.1. The grain size distribution of the initial bed is based on field data at the Lijin gauging station, and
is shown in Fig. 2. Using the sediment transport relation of Naito et al. (accepted subject to revision) for mixtures, such a grain
size distribution combined with the given bed slope and flow discharge leads to a total equilibrium sediment transport rate per
unit width $q_{seT}$ of 0.0272 m²/s. With a flood intermittency factor $I_f$ of 0.14, this further gives a mean annual bed material load
of 95.5 Mt/a. Adding in washload according to the estimate of Naito et al. (accepted subject to revision), total mean annual



load 173.7 Mt/a, a value that is of the same order of magnitude as averages over the period 2000-2016 (89-126 Mt/a depending
on site), i.e. since the operation of Xiaolangdi Dam in 1999 (Fig. 1(b)). The sediment supply rate of each grain size range is
set at 10% of its equilibrium sediment transport rate. This results in a total sediment supply rate of $q_{sf} = 0.00272$ m$^2$/s, and a
grain size distribution of the sediment supply (shown in Fig. 2) that is identical to the grain size distribution of the equilibrium
sediment load. Again we exhibit simulation results for only 0.2 year here, a value that is enough to show the differences
between the two forms, flux and entrainment, as applied to mixtures. Modeling results over a longer time scale are presented
in Section 4.3.

Figure 6 shows the simulation results using the flux form of the Exner equation. As a result of the reduced sediment

supply at the inlet, bed degradation occurs first at the upstream end and then gradually migrates downstream. The total sediment
transport rate per unit width $q_{sT}$ also reduces as a response to the cutoff of sediment supply. More specifically, the evolution
of $q_{sT}$ shows marked evidence of advection, with at least two kinematic waves being observed within 0.2 year. As shown in
Fig. 6(b), the fastest kinematic wave migrates beyond the 200 km reach within 0.06 year, and the second fastest kinematic
wave migrates for a distance of about 60 km in 0.2 year. Figures 6(c) and 6(d) show the results for the surface geometric mean
grain size $D_{sg}$ and geometric mean grain size of suspended load $D_{lg}$ respectively. As can be seen therein, both the bed surface
and the suspended load coarsen as a result of the cutoff of sediment supply. Such coarsening is not evident near the upstream
end, possibly due to the inverse slope visible in Fig. 6(a). Similarly to the variation of $q_{sT}$, the patterns of time variation of both
$D_{sg}$ and $D_{lg}$ also exhibit very clear kinematic waves, with migration rates about the same as those of $q_{sT}$.





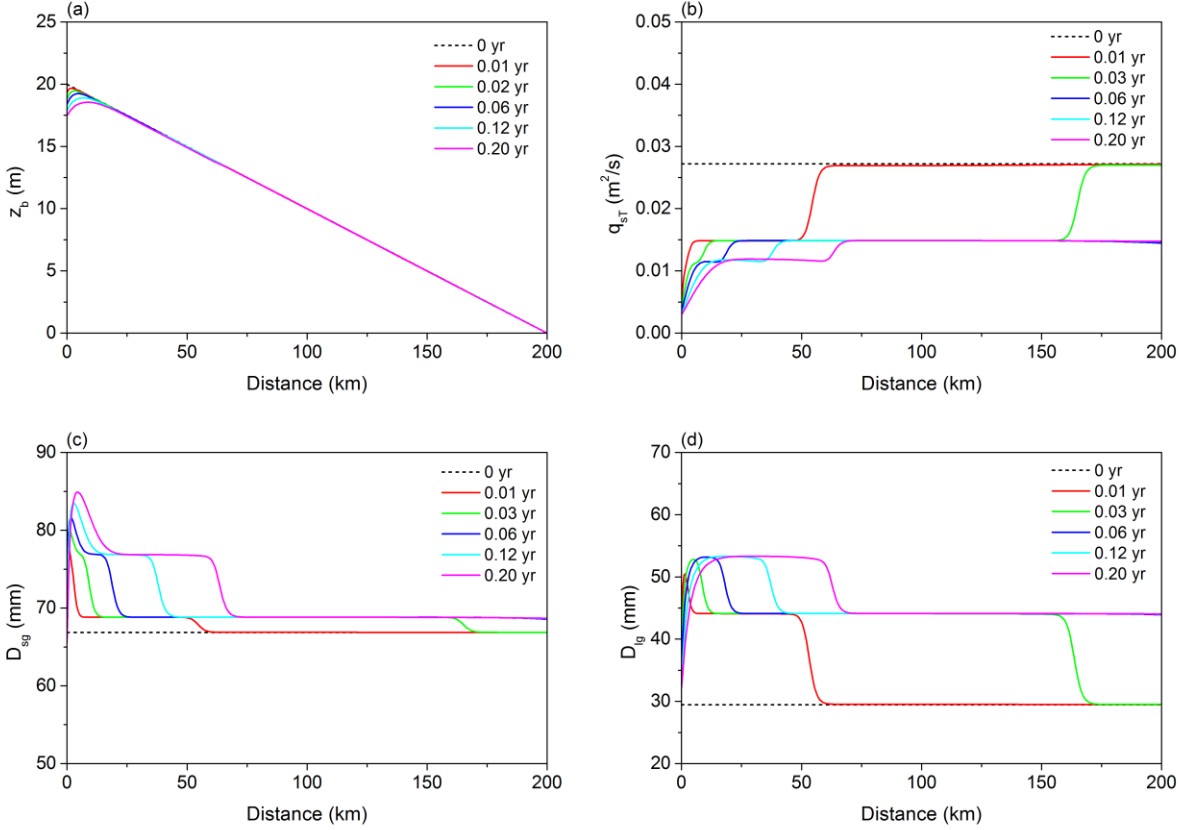

**Figure 6.** 0.2 year results for the case of sediment mixtures using the flux form of Exner equation: time variation of (a) bed elevation $z_b$, (b) total sediment load $q_{sT}$, (c) surface geometric mean grain size $D_{sg}$ and (d) geometric mean grain size of sediment load of the LYR in response to the cutoff of sediment supply.

Figure 7 shows the simulation results obtained using the entrainment form of the Exner equation. In general, the patterns of variation predicted by the entrainment form have similar trends and magnitudes to those predicted by the flux form: the bed degrades near the upstream end, the suspended load transport rate reduces in time, and both the bed surface and the suspended load coarsen as a result of the cutoff of sediment supply. But the results based on the two forms exhibit very evident differences when multiple grain sizes are included. That is, the results predicted by the entrainment form are sufficiently diffusive so that the variations of $q_{sT}$, $D_{sg}$, and $D_{lg}$ (Figs. 7(b), 7(c) and 7(d)) do not show the advective character seen in Fig. 6. No clear kinematic waves can be observed in Fig. 7. Table 3 gives a summary of the values of $\delta$ in the case of sediment mixtures. The prediction of bed elevation is not affected much when multiple grain sizes are considered, with $\delta(z_b)$ being no more than 3.5% within 0.2 year. The $\delta$ values of $q_{sT}$, $D_{sg}$, and $D_{lg}$ are, however, relatively large since the two forms predict quite different patterns of variations, as shown in Fig. 6 and Fig. 7.





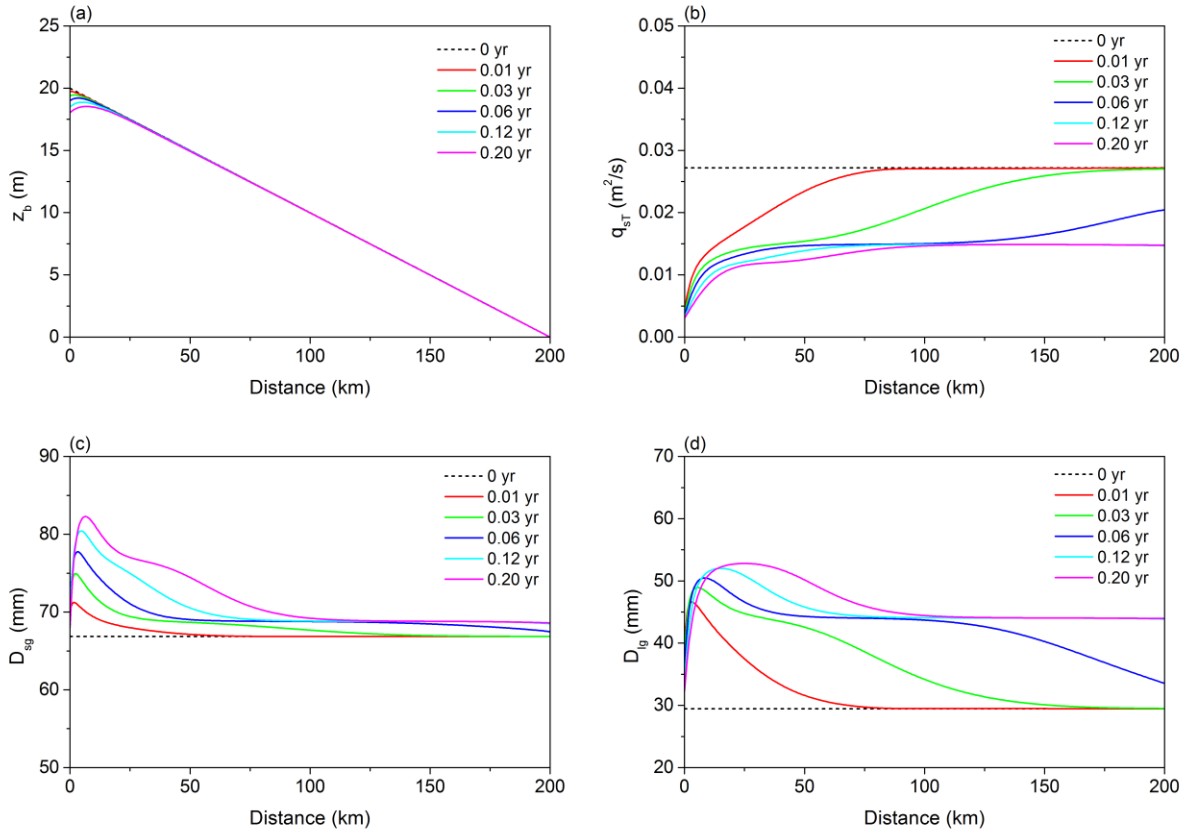

**Figure 7.** 0.2 year results for the case of sediment mixtures using the entrainment form of Exner equation: time variation of (a) bed elevation $z_b$, (b) total sediment load $q_{sT}$, (c) surface geometric mean grain size $D_{sg}$ and (d) geometric mean grain size of sediment load of the LYR in response to the cutoff of sediment supply.

**Table 3.** Quantification of the difference between predictions of the flux form and the entrainment form in the case of sediment mixtures. The maximum values of $\delta$ in the calculational domain are presented at different times.

|  |  | 0.01 yr | 0.03 yr | 0.06 yr | 0.12 yr | 0.20 yr |
|---|---|---|---|---|---|---|
| original $v_s$ | $\delta(z_b)$ | 2.31 % | 3.15 % | 3.44 % | 3.44 % | 3.24 % |
|  | $\delta(q_{sT})$ | 54.66 % | 76.11 % | 41.13 % | 10.45 % | 11.77 % |
|  | $\delta(D_{sg})$ | 10.07 % | 8.60 % | 7.18 % | 6.02 % | 5.40 % |
|  | $\delta(D_{lg})$ | 27.10 % | 31.87 % | 23.67 % | 7.16 % | 7.68 % |
| $v_s$ multiplied by 20 | $\delta(z_b)$ | 0.27 % | 0.40 % | 3.83 % | 0.26 % | 0.21 % |
|  | $\delta(q_{sT})$ | 81.12 % | 82.26 % | 39.65 % | 7.15 % | 9.28 % |
|  | $\delta(D_{sg})$ | 2.81 % | 2.84 % | 1.96 % | 2.65 % | 3.41 % |
|  | $\delta(D_{lg})$ | 32.78 % | 33.11 % | 25.13 % | 4.77 % | 6.02 % |





The results shown in Fig. 8 have also been calculated using the entrainment form of the Exner equation, but here the
sediment fall velocities $v_{si}$ used in Eqs. (14)-(16) are arbitrarily multiplied by a factor of 20. That is, we still apply the grain
size distribution in Fig. 2, but the sediment fall velocities implemented in the simulation are 20 times the corresponding fall
velocities calculated by the relation of Dietrich (1982). In the case of uniform sediment in Section 3.1, we arbitrarily reduce
the sediment fall velocity to force a difference between the predictions from the entrainment form and those from the flux form.
Here we arbitrarily increase the sediment fall velocity with the aim of determining under what conditions the sorting patterns
predicted by the two forms converge. As we can see in Fig. 8, with such a larger and intentionally unrealistic sediment fall
velocity, the general trend of variations predicted by the entrainment form does not change, but the results show a notably less
diffusive pattern. The variations of $q_{sT}$, $D_{sg}$, and $D_{lg}$ show more advection compared with Fig. 7, and at least two kinematic
waves appear within 0.2 year. It should be noted that even though these kinematic waves appear after we arbitrarily increase
the sediment fall velocity, they are more diffusive than those obtained from the flux formulation and also migrate with a slower
celerity as compared with those predicted by the flux form, especially for the fastest kinematic wave in the modeling results.
Table 3 summarizes the $\delta$ values for this run. The values of $\delta(z_b)$ become smaller with arbitrarily increased sediment
fall velocities except for $t = 0.06$ year. A relatively large value of $\delta(z_b)$ at $t = 0.06$ year occurs near the downstream end of the
channel, where the entrainment form predicts some slight degradation. Also, $\delta(q_{sT})$ is quite large at $t = 0.01$ year and 0.03 year,
even though the results for the case of increased fall velocities become qualitatively more similar to the prediction of the flux
form. This is because the flux form and the entrainment form with arbitrarily increased sediment fall velocities predict different
celerities for the fastest kinematic wave. The error $\delta(q_{sT})$ becomes smaller from $t = 0.06$ year as the fastest kinematic wave
migrates beyond the channel reach. The error $\delta(D_{lg})$ behaves similarly to $\delta(q_{sT})$, with $\delta(D_{lg})$ being quite large at $t = 0.01$ year
and 0.03 year near the fastest kinematic wave, but gradually becoming smaller as time passes. The error $\delta(D_{sg})$ stays low within
the whole 0.2 year period, possibly because the fastest kinematic wave of $D_{sg}$ has a small magnitude, as shown in Fig. 8(c).



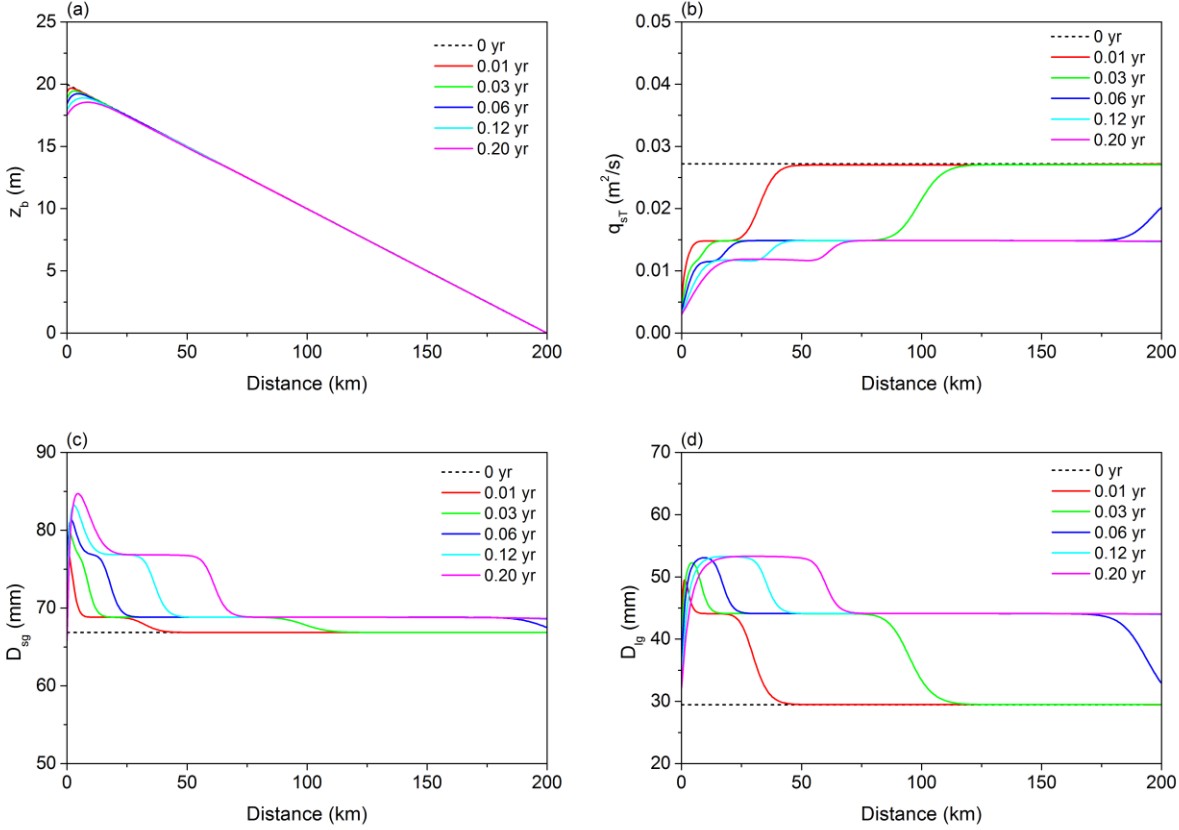

**Figure 8.** 0.2 year results for the case of sediment mixtures using the entrainment form of Exner equation: time variation of
(a) bed elevation $z_b$, (b) total sediment load $q_{sT}$, (c) surface geometric mean grain size $D_{sg}$ and (d) geometric mean grain size
of sediment load of the LYR in response to the cutoff of sediment supply. Sediment fall velocities $v_{si}$ are arbitrarily multiplied
by a factor of 20 in this run while keeping the grain sizes invariant.
**4. Discussion**
**4.1 Adjustment of sediment load and the adaptation length**
In Section 3.1, our simulation shows that in the case of uniform sediment, the flux form and the entrainment form of
the Exner equation give very similar predictions for a given sediment size of 65 μm. However, if we arbitrarily reduce the
sediment fall velocity by a multiplicative factor of 0.05, the prediction given by the entrainment form will become much more
diffusive, in terms of both $z_b$ and $q_s$. The diffusive nature of the entrainment form as well as the important role played by the
sediment fall velocity can be explained in terms of the governing equation.
In the entrainment form, the equation governing suspended sediment concentration is,





$$\frac{1}{I_f}\frac{\partial(hC)}{\partial t}+\frac{\partial(huC)}{\partial x}=v_s\left(E-r_0C\right)$$ (29)
i.e. the same as Eq. (10). The sediment transport rate per unit width $q_s = huC = q_wC$, and the dimensionless entrainment rate
$E = r_0 q_{se}/q_w$. If we consider only the adjustment of sediment concentration in space and neglect the temporal derivative in Eq.
(29), we get
$$\frac{\partial q_s}{\partial x}=v_s\left(E-r_0C\right)=\frac{1}{L_{ad}}\left(q_{se}-q_s\right)$$ (30)
$$L_{ad}=\frac{q_w}{v_s r_0}$$ (31)
where $L_{ad}$ can be identified as the adaptation length for suspended sediment to reach equilibrium. This definition of adaptation
length is similar to those in Wu and Wang (2008), and Ganti et al. (2014).
If we consider the spatial adjustment of sediment load shortly after the cutoff of sediment supply, we can further
neglect the nonuniformity of the capacity (equilibrium) transport rate $q_{se}$ along the channel, and Eq. (30) can be solved with a
given upstream boundary condition. That is, with the boundary condition
$$q_s\big|_{x=0}=q_{sf}$$ (32)
Eq. (30) can be solved to yield
$$q_s=q_{se}+\left(q_{sf}-q_{se}\right)e^{-\frac{x}{L_{ad}}}$$ (33)
Here $q_{sf}$ is the sediment supply rate per unit width at the upstream end. According to Eq. (33), $q_s$ adjusts exponentially in space
from $q_{sf}$ to $q_{se}$, which also coincides with our simulation in Section 3.1 as shown in Figs. 3-6. The adaptation length $L_{ad}$ is the
key parameter that controls the distance for $q_s$ to approach the equilibrium sediment transport rate $q_{se}$. More specifically, $q_s$
attains 1 - 1/e (i.e. 63.2%) of its adjustment from $q_{sf}$ to $q_{se}$ over a distance $L_{ad}$. Therefore, the larger the adaptation length, the
slower $q_s$ adjusts in space, so that the more evident lag effects and diffusivity are exhibited in the entrainment form. In the flux
form, however, the sediment load responds simultaneously with the flow conditions, so that $L_{ad} = 0$ and $q_s = q_{se}$ along the
entire channel reach.
For the case of uniform sediment in Section 3.1, $q_w = 6.67$ m$^2$/s and $r_o$ is specified as unity. Therefore, the value of
$L_{ad}$ is determined only by the sediment fall velocity $v_s$. Figure 9 shows the value of the adaptation length $L_{ad}$ for various
sediment grain sizes, with the sediment fall velocity $v_s$ calculated by the relation of Dietrich (1982). From the figure we can





see that $L_{ad}$ decreases sharply with the increase of grain size, indicating that the lag effects between sediment transport and
flow conditions are evident for very fine sediment but gradually disappear when sediment is sufficiently coarse. For the
sediment grain size of 65 μm implemented in Section 3.1, the corresponding $L_{ad} = 1.88$ km, which is much smaller than the
200 km reach of the computational domain. Therefore, the predictions of the flux form and the entrainment form show little
difference. However, if we arbitrarily multiply the sediment fall velocity by a factor of 0.05, then $L_{ad}$ becomes 37.60 km. With
such a large adaptation length, it is no surprise that the entrainment form gives very different prediction from the flux form.

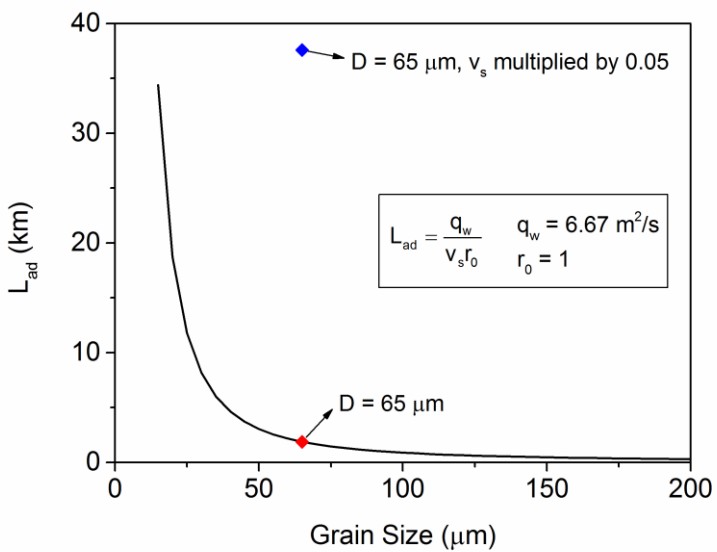

**Figure 9.** Relation between adaptation length $L_{ad}$ and grain size $D$. The values of flow discharge per unit width $q_w$ and recovery
coefficient $r_0$ are the same as those in Section 3.1. The relation of Dietrich (1982) is implemented for sediment fall velocity.

The evolution of bed elevation $z_b$ can also be affected by the value of $L_{ad}$. For example in the case of uniform sediment

in Section 3.1, the flux form corresponds to an adaption length of zero. As a result, the flux form yields a spatial derivative of
$q_s$ near the upstream end that is relatively large, thus leading to fast degradation from the upstream end. In the case of the
entrainment form, however, the spatial derivative of $q_s$ is small with a large $L_{ad}$, thus leading to a slower and more diffusive
bed degradation. This is especially evident when we arbitrarily reduce the sediment fall velocity by a factor of 0.05, while
keeping grain size invariant.

The above analysis also holds for sediment mixtures, except that each grain size range will have its own adaptation

length. Here we neglect the temporal derivative in Eq. (29) and analyze only the spatial adjustment of sediment load. If we
neglect the spatial derivative in Eq. (29) and conduct a similar analysis for sediment concentration, we would find that the
temporal adjustment of sediment concentration is also described by an exponential function of time, in analogy to Eq. (33).





## 4.2 Patterns of grain sorting: advection vs. diffusion

In Section 3.2 we find that the flux form and entrainment form of the Exner equation provide very different patterns of grain sorting for sediment mixtures: kinematic sorting waves are evident in the flux form but are diffused out in the entrainment form. The diffusivity of grain sorting becomes smaller and the kinematic waves appear, however, if we arbitrarily increase the sediment fall velocity by a factor of 20. In this section, we explain this behavior by analyzing the governing equations.

First we rewrite the sediment transport relation of Naito et al. (accepted subject to revision) in the following form,

$$q_{sei} = F_i q_{ri} \tag{34}$$

$$q_{ri} = \frac{u_*^3}{RgC_f} A_i \left( \tau_g^* \frac{D_g}{D_i} \right)^{B_i} \tag{35}$$

Substituting Eq. (34) into Eq. (6), which is the governing equation for surface fraction $F_i$ in the flux form, we get

$$\frac{1}{I_f} \left(1 - \lambda_p\right) \left[ L_a \frac{\partial F_i}{\partial t} + \left(F_i - f_{Ii}\right) \frac{\partial L_a}{\partial t} \right] = f_{Ii} \frac{\partial \sum_{j=1}^{n} F_j q_{rj}}{\partial x} - \frac{\partial F_i q_{ri}}{\partial x} \tag{36}$$

Equation (36) can be written in the form of a kinematic wave equation with source terms as below,

$$\frac{\partial F_i}{\partial t} + c_{Fi} \frac{\partial F_i}{\partial x} = SF_i \tag{37}$$

$$c_{Fi} = \frac{I_f q_{ri}}{\left(1 - \lambda_p\right) L_a} \left(1 - f_{Ii}\right) \tag{38}$$

$$SF_i = -\frac{I_f F_i \left(1 - f_{Ii}\right)}{\left(1 - \lambda_p\right) L_a} \frac{\partial q_{ri}}{\partial x} + \frac{I_f f_{Ii}}{\left(1 - \lambda_p\right) L_a} \frac{\partial \sum_{j=1}^{n, j \neq i} F_j q_{rj}}{\partial x} - \frac{F_i - f_{Ii}}{1 - \lambda_p} \frac{\partial L_a}{\partial t} \tag{39}$$

where $c_{Fi}$ is the $i$-th celerity of kinematic wave and $SF_i$ denotes source terms. Since the surface geometric mean grain size $D_{sg}$, the total sediment load per unit width $q_{sT}$ (which equals the equilibrium sediment transport rate $q_{seT}$), and the geometric mean grain size of sediment load $D_{lg}$ are all closely related to the surface grain size fractions $F_i$, the evolution of these three





parameters exhibit marked advective behavior when simulated by the flux form of the Exner equation. However, the evolution
of bed elevation $z_b$ is related with $\partial q_{sT}/\partial x$, which is dominated by diffusion if $q_{sT}$ is predominantly slope-dependent.
Now we turn to the entrainment form of the Exner equation. Combined with the sediment transport rate per unit width
$q_{si} = huC_i = q_wC_i$ and the dimensionless entrainment rate $E_i = r_{0i}q_{sei}/q_w$, Eq. (16) and Eq. (15) can be written as,
$$\frac{1}{I_f}\frac{\partial\left(\frac{q_{si}}{u}\right)}{\partial t} + \frac{\partial q_{si}}{\partial x} = \frac{v_{si}r_{oi}}{q_w}\left(q_{sei} - q_{si}\right) \tag{40}$$
$$\frac{1}{I_f}\left(1-\lambda_p\right)\left[L_a\frac{\partial F_i}{\partial t} + \left(F_i - f_{Ii}\right)\frac{\partial L_a}{\partial t}\right] = f_{Ii}\sum_{j=1}^{n}\frac{v_{sj}r_{0j}}{q_w}\left(q_{sej} - q_{sj}\right) - \frac{v_{si}r_{0i}}{q_w}\left(q_{sei} - q_{si}\right) \tag{41}$$
where Eq. (40) denotes the conservation of suspended sediment and Eq. (41) denotes the conservation of bed material. If we
rewrite Eq. (40) in the following form,
$$q_{si} = q_{sei} - \frac{q_w}{v_{si}r_{oi}}\left[\frac{1}{I_f}\frac{\partial\left(\frac{q_{si}}{u}\right)}{\partial t} + \frac{\partial q_{si}}{\partial x}\right] \tag{42}$$
then $q_{si}$ can be solved iteratively. With an initial guess of $q_{si} = q_{sei}$ and neglecting the temporal derivatives, we obtain the second
order solution of $q_{si}$ as,
$$q_{si} = q_{sei} - \frac{q_w}{v_{si}r_{oi}}\frac{\partial}{\partial x}\left(q_{sei} - \frac{q_w}{v_{si}r_{oi}}\frac{\partial q_{sei}}{\partial x}\right) \tag{43}$$
Details of the iteration are given in Appendix B.
Substituting Eq. (43) and Eq. (34) into Eq. (41), we find that
$$\frac{1}{I_f}\left(1-\lambda_p\right)\left[L_a\frac{\partial F_i}{\partial t} + \left(F_i - f_{Ii}\right)\frac{\partial L_a}{\partial t}\right] = f_{Ii}\sum_{j=1}^{n}\frac{\partial}{\partial x}\left(F_jq_{rj} - \frac{q_w}{v_{sj}r_{oj}}\frac{\partial F_jq_{rj}}{\partial x}\right) - \frac{\partial}{\partial x}\left(F_iq_{ri} - \frac{q_w}{v_{si}r_{oi}}\frac{\partial F_iq_{ri}}{\partial x}\right) \tag{44}$$
Expanding out the last two terms in Eq. (44) using the chain rule, after some work the relation for the conservation of bed
material can be expressed as,




$$\frac{\partial F_i}{\partial t} + c_{Ei}\frac{\partial F_i}{\partial x} - v_i\frac{\partial^2 F_i}{\partial x^2} = SE_i \qquad (45)$$
$$c_{Ei} = \frac{(1-f_{Ii})I_f}{(1-\lambda_p)L_a}\left(q_{ri} - 2\frac{q_w}{v_{si}r_{0i}}\frac{\partial q_{ri}}{\partial x}\right) \qquad (46)$$
$$v_i = \frac{(1-f_{Ii})I_f q_w q_{ri}}{(1-\lambda_p)L_a v_{si} r_{0i}} \qquad (47)$$
$$SE_i = \frac{I_f f_{Ii}}{(1-\lambda_p)L_a}\sum_{j=1}^{n,j\neq i}\frac{\partial}{\partial x}\left(F_j q_{rj} - \frac{q_w}{v_{sj}r_{oj}}\frac{\partial F_j q_{rj}}{\partial x}\right) - \frac{(1-f_{Ii})I_f}{(1-\lambda_p)L_a}\left(F_i\frac{\partial q_{ri}}{\partial x} - \frac{q_w}{v_{si}r_{oi}}F_i\frac{\partial^2 q_{ri}}{\partial x^2}\right) - \frac{F_i - f_{Ii}}{L_a}\frac{\partial L_a}{\partial t} \qquad (48)$$
where $c_{Ei}$ is the celerity of kinematic wave, $v_i$ is the diffusivity coefficient, and $SE_i$ denote source terms.
From Eq. (45) we can see that the governing equation for $F_i$ in the entrainment form is an advection-diffusion equation,
rather than the kinematic wave equation of the flux form. The surface geometric mean grain size $D_{sg}$ is governed by Eq. (45),
with describes the variation of the surface fractions $F_i$ from which it is computed. The equilibrium sediment transport rate $q_{sei}$
is governed by Eq. (45) because we implement a surface-based sediment transport relation as shown in Eq. (34). According to
Eq. (43), the total sediment load per unit width $q_{sT}$ and the geometric mean grain size of sediment load $D_{lg}$ must also be closely
related to the surface grain size fractions $F_i$. Therefore, the diffusion terms in Eq. (45) can lead to the dissipation of the
kinematic waves in Figs. 7(b), 7(c), and 7(d).
From Eq. (47), we can also see that the diffusivity coefficient $v_i$ is related to the sediment fall velocity $v_{si}$: the larger
the sediment fall velocity, the smaller the diffusivity coefficient. Thus when we increase the sediment fall velocity arbitrarily
by a factor of 20 in Section 3.2, the kinematic waves become more evident as a result of the reduction of diffusivity.
Moreover if we compare the celerity of kinematic waves in both the flux form and the entrainment form, we have
$$\frac{c_{Ei}}{c_{Fi}} = 1 - r_{ci} \qquad (49)$$
$$r_{ci} = 2\frac{L_{adi}}{q_{ri}}\frac{\partial q_{ri}}{\partial x} \qquad (50)$$
where $L_{adi}$ is the adaptation length for the $i$-th size range as defined by Eq. (31). More specifically, the value of $r_{ci}$ depends on
$\partial q_{ri}/\partial x$. For our numerical simulation in Section 3.2, $\partial q_{ri}/\partial x > 0$ as a result of bed degradation progressing from the upstream
end, thus leading to a positive value of $r_{ci}$ and an entrainment celerity $c_{Ei}$ that is smaller than the corresponding flux celerity




$c_{Fi}$. This is consistent with our numerical results: the kinematic waves in Fig. 8 predicted by the entrainment form are somewhat
smaller than the kinematic waves in Fig. 6 predicted by the flux form.

**4.3 Modeling implications and limitations**

In Section 3, two numerical cases are conducted to compare the flux form and the entrainment form of the Exner
equation, but only within 0.2 year after the cutoff of sediment supply. Here we run both numerical cases for a longer time (5
years). Table 4 shows the results of the case of uniform sediment (as described in Section 3.1) within 5 years, and Table 5
shows the results of the case of sediment mixtures (as described in Section 3.2) within 5 years. For both cases, the $\delta$ values,
corresponding to relative deviation between the flux and entrainment forms, become quite small after 1 year, thus validating
our assumption that the predictions of the two forms tend to be most evident shortly after disruption, but gradually diminish
over a longer time scale. Moreover, if the water and sediment supply are kept constant for a sufficiently long time, the flux
form and entrainment form of Exner equation predict exactly the same equilibrium, under which the equilibrium sediment
transport rate (of each size range) equals to the sediment supply rate (of each size range).
**Table 4.** Quantification of the difference between predictions of the flux form and the entrainment form in the case of uniform
sediment. The maximum $\delta$ in the calculational domain are presented for each of 5 years.

|  |  | 1 yr | 2 yr | 3 yr | 4 yr | 5 yr |
|---|---|---|---|---|---|---|
| original $v_s$ | $\delta(z_b)$ | 2.97 % | 2.67 % | 2.56 % | 2.54 % | 2.55 % |
|  | $\delta(q_s)$ | 2.97 % | 1.77 % | 1.31 % | 1.09 % | 1.00 % |

**Table 5.** Quantification of the difference between predictions of the flux form and the entrainment form in the case of sediment
mixtures. The maximum $\delta$ in the calculational domain are presented for each of five years.

|  |  | 1 yr | 2 yr | 3 yr | 4 yr | 5 yr |
|---|---|---|---|---|---|---|
| original $v_s$ | $\delta(z_b)$ | 2.16 % | 1.85 % | 1.74 % | 1.70 % | 1.71 % |
|  | $\delta(q_{sT})$ | 2.90 % | 1.84 % | 1.51 % | 1.40 % | 3.89 % |
|  | $\delta(D_{sg})$ | 5.22 % | 3.93 % | 3.54 % | 4.74 % | 3.92 % |
|  | $\delta(D_{lg})$ | 0.76 % | 0.61 % | 0.96 % | 1.34 % | 0.82 % |

Based on the numerical modeling and mathematical analysis in this paper, we summarize below the circumstances
under which the entrainment form of the Exner equation might be required. (1) The difference in the predictions of the two
forms of the Exner equation tends to be large shortly after disruption, but gradually diminishes over time. Therefore, we suggest
that the entrainment form of Exner equation should be used at short time scale (e.g., within a flood event), but that the flux
form of Exner equation is applicable to long-term river morphodynamics (e.g., more than one year). (2) The entrainment form
of the Exner equation is necessary if sorting processes are to be studied. The flux form of Exner equation cannot consider the



lag effects and diffusivity of individual size fractions, and therefore will result in an overestimation of the effect of advection
on sorting processes. (3) The entrainment form of the Exner equation is necessary when dealing with fine-grained sediment
(or more specifically sediment with small fall velocity), since the adaptation length $L_a$ and the diffusivity coefficient $\nu_i$ are
large under such circumstances. The flux form of the Exner equation is particularly applicable for coarse sediment, or when
the sediment transport is dominated by bedload (e.g. gravel-bed rivers). The above results could have practical implications in
regard to a wide range of issues including dam construction, water and sediment regulation, flood management, and ecological
restoration schemes. The results can also be used as a reference for other fine grained fluvial systems similar to the LYR, such
as the Pilcomayo River in Paraguay/Argentina, South America (Martń-Vide et al., 2014).

It should be noted that in the morphodynamic models of this paper, we implement the mass and momentum

conservation equations for clear water (i.e., Eq. (1) and Eq. (2)) to calculate flow hydraulics, instead of the mass and momentum
equations for water-sediment mixture as suggested by Cao et al. (2004) and Cao et al. (2006). More specifically, Cui et al.
(2005) have pointed out that when sediment concentration in the water is sufficiently small, bed elevation can be taken to be
unchanging over characteristic hydraulic time scales, and the effects of flow-bed exchange on flow hydraulics can be neglected.
For the two simulation cases in this paper, the volume sediment concentration $C$ drops from about $2 \times 10^{-3}$ to about $2 \times 10^{-4}$ in
the case of uniform sediment, and from about $4 \times 10^{-3}$ to about $4 \times 10^{-4}$ in the case of sediment mixtures, due to the cutoff of
sediment supply at the upstream end. These dilute concentrations validate our implementation of mass and momentum
conservation equations for clear water. Our assumption is not necessarily correct for the entire Yellow River. Upstream of our
study reach, and especially upstream of Sanmenxia Dam, the flow is often hyperconcentrated (Xu, 1999).

Considering the fact that in our numerical simulations a constant inflow discharge (along with a flood intermittency

factor) is implemented, and also considering that the morphodynamic time scale is much larger than the hydraulic time scale
in our case, the quasi-steady approximation or even the normal flow approximation can be introduced to further save
computational efforts (Parker, 2004). But one thing that should be noted is that in our simulation results in Section 3, the bed
exhibits an inverse slope near the upstream end. The normal flow assumption becomes invalid under such circumstances, so
requiring a full unsteady shallow water model.

By definition, the recovery coefficient $r_o$ is the ratio of the near-bed to the flux-depth-averaged concentration of

suspended load, and is thus related to the concentration profile. In our simulation $r_0$ is specified as unity. That is, density
stratification effects of suspended sediment are neglected, and the vertical profile of sediment concentration is regarded as
uniform. However in natural rivers, the value of $r_0$ can vary significantly under different circumstances (Cao et al., 2004; Duan
and Nanda, 2006; Zhang and Duan, 2011; Zhang et al., 2013). In general, the value of $r_0$ is no less than unity and can be as
large as 12 (Zhang and Duan, 2011). Therefore according to our mathematical analysis in Section 4.1 and 4.2, $r_0 = 1$
corresponds to a maximum adaptation length $L_{ad}$, a maximum diffusivity coefficient $\nu_i$, and a minimum ratio of celerities $c_{Ei}/c_{Fi}$,
thus leading to the largest difference between the flux form and the entrainment form. When sediment concentration is
sufficiently high, hindered settling effects reduce the sediment fall velocity. Considering the fact that the sediment





concentrations considered in our simulation are fairly small, hindered effects are not likely significant. More study on
stratification and hindered settling effects are merited in the case of the LYR.
**5 Conclusion**
In this paper, we compare two formulations for sediment mass conservation in context of the Lower Yellow River,
i.e. the flux form of Exner equation and the entrainment form of Exner equation. In the flux form of the Exner equation, the
conservation of bed material is related to the streamwise gradient of sediment transport rate, which is in turn computed based
on the quasi-equilibrium assumption according to which the local sediment transport rate equals the capacity rate. In the
entrainment form of the Exner equation, on the other hand, the conservation of bed material is related to the difference between
the entrainment rate of sediment from the bed into the flow and the deposition rate of sediment from the flow onto the bed. A
nonequilibrium sediment transport formulation is applied, so that the sediment transport rate can lag in space and time behind
changing flow conditions. Despite the fact that the entrainment form is usually recommended for the morphodynamic modeling
of the LYR due to its fine-grained sediment, there has been little discussion of the differences in predictions between the two
forms.
Here we implement a 1-D morphodynamic model for this problem. The fully unsteady Saint Venant Equations are
implemented for the hydraulic calculation. Both the flux form and the entrainment form of Exner equation are implemented
for sediment conservation. For each formulation, we include the options of both uniform sediment and sediment mixtures.
Two generalized versions of the Engelund-Hansen relation specifically designed for the LYR are implemented to calculate the
quasi-equilibrium sediment transport rate (i.e., sediment transport capacity). They are the version of Ma et al. (2017) for
uniform sediment, and the version of Naito et al. (accepted subject to revision) for sediment mixtures. The method of Viparelli
et al. (2010) is implemented to store and access bed stratigraphy as the bed aggrades and degrades. We apply the
morphodynamic model to two cases with conditions typical of the LYR.
In the first case, a uniform bed material grain size of 65 μm is implemented. We study the effect of cutoff of sediment
supply, as occurred after the operation of Xiaolangdi Dam in 1999. We find that the flux form and the entrainment form give
very similar predictions for this case. Through quantification of the difference between the two forms with a normalized
measure of relative difference, we find that difference in the prediction of bed elevation is quite small ($< 4\%$), but difference
in the prediction of sediment load can be relatively large (about 20%) shortly after the cutoff of sediment supply. Moreover,
the predictions of the entrainment form become very different from those of the flux form if we arbitrarily reduce the sediment
fall velocity by a multiplicative factor of 0.05, while keeping grain size unchanged.
The results for the case of uniform sediment can be explained by analyzing the governing equation of sediment load
$q_s$. In the flux form, the volume sediment transport rate per unit width $q_s$ = the local equilibrium (capacity) value $q_{se}$. But in
the entrainment form, we find that the difference between $q_s$ and $q_{se}$ decays exponentially in space. The adaptation length $L_{ad}$
= $q_w / (v_s\, r_0)$ is the key parameter that controls the distance for $q_s$ to approach its equilibrium value $q_{se}$. The larger the adaptation





length, the more different the predictions of the two forms will be. For computational conditions in this case, the adaption length is relatively small ($L_{ad}$ = 1.88 km), but it becomes much larger ($L_{ad}$ = 37.6 km) if sediment fall velocity is arbitrarily divided by a factor of 20.

In the second case the bed material consists of mixtures ranging from 15 μm to 500 μm. We find that the flux form and the entrainment form give very different patterns of grain sorting. Evident kinematic waves occur at various timescales in the flux form, but no evident kinematic waves can be observed in the entrainment form. The different sorting patterns are reflected in the evolution of surface geometric mean grain size $D_{sg}$, total sediment load $q_{sT}$ and geometric mean grain size of sediment load $D_{lg}$, but are not reflected in the evolution of bed elevation $z_b$. Kinematic waves appear in the entrainment form if we arbitrarily increase the sediment fall velocity by multiplying a factor of 20 without changing the grain size. This large increase in fall velocity leads to a large decrease in adaptation length, so that the entrainment form behaves much more like the flux form. This notwithstanding, the kinematic waves are still more diffusive and slower than those predicted by the flux form.

The different sorting patterns exhibited in the case of sediment mixtures can be explained by analyzing the governing equation for bed surface fractions $F_i$, i.e. the grain size-specific conservation of bed material. We find that in the flux form, the governing equation for $F_i$ can be written in the form of a kinematic wave equation. In the entrainment form, however, the governing equation for $F_i$ is an advection-diffusion equation. It is the diffusion term which can lead to the dissipation of kinematic waves. Moreover, in the advection-diffusion equation arising from the entrainment form, the coefficient of diffusivity is inversely proportional to the sediment fall velocity. In addition, under the condition of bed degradation the wave celerity is smaller than the celerity arising from the flux form.

Overall, our results indicate that the more complex entrainment form of the Exner equation might be required under the following circumstances: (1) when short-term (e.g., within a flood event) river morphodynamics is considered; (2) when sorting processes are studied; and (3) when fine-grained sediment (or more specifically sediment with small fall velocity) is considered.

**Appendix A: Comparison of two relations for sediment fall velocity: Dietrich (1982) against Ferguson and Church (2004)**

In this paper, we implement the relation of Dietrich (1982) to calculate sediment fall velocity $v_s$. The relation is,

$$v_s = R_f \sqrt{RgD} \tag{A1}$$

$$\ell n\left(R_f\right) = -b_1 + b_2 \ell n\left(\text{Re}_p\right) - b_3\left[\ell n\left(\text{Re}_p\right)\right]^2 - b_4\left[\ell n\left(\text{Re}_p\right)\right]^3 + b_5\left[\ell n\left(\text{Re}_p\right)\right]^4 \tag{A2}$$





$$\text{Re}_p = \frac{\sqrt{RgD}D}{\nu} \qquad\qquad (A3)$$

where $b_1 = 2.891394$, $b_2 = 0.95296$, $b_3 = 0.056835$, $b_4 = 0.002892$, $b_5 = 0.000245$, and $\nu = 10^{-6}$ is the kinematic viscosity of water.

Another widely used relation for sediment fall velocity is the relation of Ferguson and Church (2004), which is regarded as applying to nearly the entire range of viscous to turbulent conditions.

$$v_s = \frac{RgD^2}{C_1\nu + \left(0.75C_2RgD^3\right)^{0.5}} \qquad\qquad (A4)$$

where $C_1 = 18$ and $C_2 = 0.4$ for smooth spheres; $C_1 = 18$ and $C_2 = 1.0$ for sieve diameters of natural sand; and $C_1 = 20$ and $C_2 = 1.1$ for nominal diameters of natural sand. More specifically, the relation of Ferguson and Church (2004) converges on Stokes' law for small grains, and to a constant drag coefficient for large grains.

Considering the fact that the sediment of LYR is finer than most sand-bed rivers (Ma et al., 2017), here we compare the two relations for sediment fall velocity in the context of the LYR. The two parameters in Ferguson and Church are specified as $C_1 = 18$ and $C_2 = 1.0$. In our simulation, the sediment size range of the LYR is specified as 15 μm ~ 500 μm.

According to Fig. A1, the relation of Dietrich (1982) and the relation of Ferguson and Church (2004) coincide with each other within this size range, thus justifying our implementation of Dietrich (1982) in the simulation. For grain sizes smaller than 15 μm, sediment becomes washload in the LYR and Dietrich (1982) predicts sediment fall velocities that are smaller than those predicted by Ferguson and Church (2004). For sediment coarser than 500 μm, Dietrich (1982) somewhat overestimates sediment fall velocity compared with Ferguson and Church (2004).



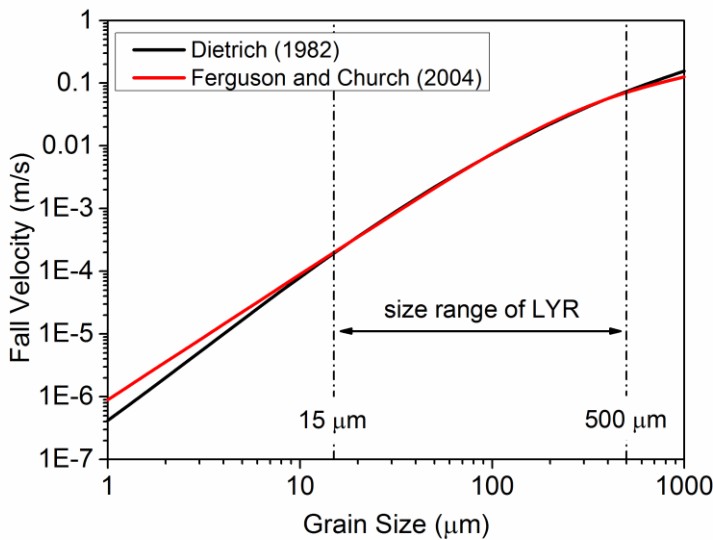

**Figure A1.** Comparison of two relations for sediment fall velocity: Dietrich (1982) and Ferguson and Church (2004)
**Appendix B: Iterative solution of sediment transport rate $q_{si}$ in the entrainment form**

The parameter $q_{si}$ in Eq. (40) is solved iteratively as below,

$$q_{si}^{(m+1)} = q_{sei} - \frac{q_w}{v_{si}r_{oi}}\left[\frac{1}{I_f}\frac{\partial\left(\frac{q_{si}^{(m)}}{u}\right)}{\partial t} + \frac{\partial q_{si}^{(m)}}{\partial x}\right]$$
(B1)

where the superscript denotes the order of iteration. The following zero-order solution is specified as an initial value;
$$q_{si}^{(0)} = q_{sei}$$
(B2)

From this we can get the first order and the second order solution,
$$q_{si}^{(1)} = q_{sei} - \frac{q_w}{v_{si}r_{oi}}\left[\frac{1}{I_f}\frac{\partial\left(\frac{q_{sei}}{u}\right)}{\partial t} + \frac{\partial q_{sei}}{\partial x}\right]$$
(B3)





$$q_{si}^{(2)} = q_{sei} - \frac{q_w}{v_{si}r_{oi}} \frac{1}{I_f} \frac{\partial}{\partial t} \frac{1}{u} \left\{ q_{sei} - \frac{q_w}{v_{si}r_{oi}} \left[ \frac{1}{I_f} \frac{\partial \left( \frac{q_{sei}}{u} \right)}{\partial t} + \frac{\partial q_{sei}}{\partial x} \right] \right\} - \frac{q_w}{v_{si}r_{oi}} \frac{\partial}{\partial x} \left\{ q_{sei} - \frac{q_w}{v_{si}r_{oi}} \left[ \frac{1}{I_f} \frac{\partial \left( \frac{q_{sei}}{u} \right)}{\partial t} + \frac{\partial q_{sei}}{\partial x} \right] \right\}$$

(B4)

The second order iterative solution in Eq. (B4) is tedious in form, but the only terms of importance on the right-hand

side are the spatial derivatives. Therefore we drop the time derivatives for simplicity. This gives,
$$q_{si} = q_{sei} - \frac{q_w}{v_{si}r_{oi}} \frac{\partial}{\partial x} \left( q_{sei} - \frac{q_w}{v_{si}r_{oi}} \frac{\partial q_{sei}}{\partial x} \right)$$                    (B5)
which corresponds to Eq. (41) as implemented in Section 4.2.
**Notation**
$C$ depth-flux-averaged sediment concentration
$C_f$ dimensionless bed resistance coefficient
$C_z$ dimensionless Chezy resistance coefficient
$c_b$ near-bed sediment concentration
$c_E$ celerity of the kinematic wave corresponding to $F_i$ in the entrainment form
$c_{Fi}$ celerity of the kinematic wave corresponding to $F_i$ in the flux form
$D$ sediment grain size
$E$ dimensionless entrainment rate of sediment
$F_i$ volumetric fraction of surface material in the $i$-th size range
$f_{Ii}$ volumetric fraction of sediment in the $i$-th size range exchanged across the surface-substrate interface
$g$ gravitational acceleration
$h$ water depth
$I_f$ flood intermittency factor
$L_a$ thickness of active layer
$L_{ad}$ adaptation length of suspended load
$p_{si}$ volumetric fraction of bed material load in the $i$-th size range
$q_{ri}$ normalized sediment transport rate per unit width for the $i$-th size range, defined by Eq. (34)
$q_s$ volumetric sediment transport rate per unit width





$q_{se}$ equilibrium volumetric sediment transport rate (capacity) per unit width
$q_{sf}$ sediment supply rate per unit width
$q_w$ flow discharge per unit width
$R$ submerged specific gravity of sediment
$r_0$ user-specified parameter denoting the ratio between the near-bed sediment concentration and the flux-averaged sediment
concentration
$S$ bed slope
$t$ time
$u$ depth-averaged flow velocity
$u_*$ shear velocity
$v_s$ sediment fall velocity
$x$ streamwise coordinate
$z_b$ bed elevation
$\alpha$ coefficient in Eq. (6) for interfacial exchange fractions
$\Delta t_h$ time step for hydraulic calculation
$\Delta t_m$ time step for morphologic calculation
$\Delta x$ spatial step length.
$\delta$ normalized parameter quantifying the fraction difference between the entrainment form and the flux form.
$\lambda_p$ porosity of bed deposit
$v_i$ diffusivity coefficient corresponding to $F_i$ in the entrainment form;
$\rho$ density of water
$\rho_s$ density of sediment
$\tau_b$ bed shear stress
$\tau^*$ dimensionless shear stress (Shields number)
**Competing interests**
The authors declare that they have no conflict of interest.
**Acknowledgments**
The participation of Chenge An and Xudong Fu was made possible in part by grants from the National Natural Science
Foundation of China (grants 51525901 and 91747207), and the Ministry of Science and Technology of China (grant
2016YFC0402406). The participation of Andrew J. Moodie, Hongbo Ma, Kensuke Naito, and Gary Parker were made possible



in part by grants from National Science Foundation (grant EAR-1427262). The participation of Yuanfeng Zhang was made
possible in part by grant from the National Natural Science Foundation of China (grant 51379087). Part of this research was
accomplished during Chenge An's visit in the University of Illinois at Urbana-Champain, which was supported by the China
Scholarship Council (file no. 201506210320). The participation of Andrew J. Moodie was also supported by an National
Science Foundation Graduate Research Fellowship (grant 145068). We thank the Morphodynamics Class of 2016 at the
University of Illinois at Urbana-Champaign for their participation in preliminary modeling efforts.

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
