# Peer review of "Morphodynamic model of Lower Yellow River: flux or entrainment form"

_Earth Surface Dynamics, 2018_

## Referee Comment (RC1) · Anonymous Referee #1 · 14 Jul 2018

Review of "Morphodynamic model of Lower Yellow River: flux or entrainment form for sediment mass conservation?" by Chenge An et al.

The authors asses the differences between modelling sediment conservation using a flux-based formulation and an entrainment-deposition type formulation. They apply the two types of models to the Lower Yellow River and study the differences between them. The writing is clear and the manuscript is well structured, although the authors could use less words for what they want to say. The analysis is definitely of interest to the ESurf reader, but I think the manuscript message could be stronger. I have two main problems with the manuscript. The first is the fact that it seems that the main ques-

tion the authors are answering ("what are the differences between an entrainment form conservation equation and when to use a flux-based formulation") rather is "When does a modeller need to properly account for the lag of suspended load tranpsport?". The second problem is the fact that the authors treat the flux form of the mass conservation equation as a synonym to a capacity-based or equilibrium approach, and the entrainment form of the mass conservation equation is considered a noncapacity-based or non-equilibrium based formulation. I think the two types of models (flux form and entrainment form) are not synonyms for capacity-based and noncapacity-based. The terms are not equivalent, which has some impact on the message of the manuscript. Also, I would suggest the authors to develop generic guidelines for when to use an entrainment form conservation equation and when to use a flux-based formulation, and only after this move toward the case of the Lower Yellow River.

Main comments: 1. It seems that the main question the authors are answering ("what are the differences between an entrainment form conservation equation and when to use a flux-based formulation") rather is "When does a modeller need to properly account for suspended load mechanisms?"

2. The authors treat the flux form of the mass conservation equation as a synonym to a capacity-based or equilibrium approach, and the entrainment form of the mass conservation equation is considered noncapacity-based or non-equilibrium. I think the two types of models (flux form and entrainment form) are not synonyms for capacity-based and noncapacity-based. These terms are not equivalent. Revising this may impact several parts of the text.

3. Associated with the previous comment: Could Bell and Sutherland (1983) and Armanini and Di Silvio (1988) be examples of the flux-form combined with a non-capacity approach?

4. Associated with the same comment: And may Blom and Parker (2004), Blom et al. (2006, 2008) be an example of the entrainment form combined with a capacity-based

approach?

5. Isn't the Exner equation of conservation of sediment mass a flux-based approach? Shouldn't the authors consider another (other than "Exner") name for the entrainment form of conservation of sediment mass?

6. Ln 26 "study this problem by comparing the results of flux-based and entrainment-based morphodynamics". Please explain in what way you are comparing the two types of conservation models. This also holds for ln 105-106. One of the ways seems to be by mimicking "the reduction of the sediment load in the LYR in recent years" (Ln 279). I think this information needs to be moved to the Introduction section. Figs 6 and 7. Shouldn't the equilibrium channel slope and bed surface texture be equal for the flux form and the entrainment form? It would be nice if the authors would confirm this.

7. Ln 26 "study this problem by comparing the results of flux-based and entrainment-based morphodynamics under conditions typical of the Lower Yellow River". I would suggest to develop generic guidelines for when to use an entrainment form conservation equation and when to use a flux-based formulation, and only after this move toward the case of the Lower Yellow River.

8. Ln 45-47. I'd rephrase this as application of the entrainment form of the conservation equation is not necessarily limited to suspended sediment.

9. Ln 57-71. These lag issues are not only covered by the entrainment form, but may also be covered by the flux form. . . Also see comment 2.

10. Ln 78. "More recently, however, since the operation of Xiaolangdi Dam in 1999 the LYR has seen a substantial reduction in its sediment load (Fig. 1(b))". - The authors do not address the distinct temporal decrease of the annual sediment load between 1950-2000. - Why is the year 2000 indicated in Figure 1b and not 1999? Please indicate in the figure whether 2000 is supposed to indicate the year of the Xiaolangdi Dam construction.

11. Figure 1 shows a number of very interesting features that are currently not addressed by the authors: - the distinct temporal decrease of the annual sediment load between 1950-2000. What is the cause of this decrease? - The three lines for the three cities are highly correlated. Please indicate and explain. - The suspended load is significantly finer than the bed material (i.e., grain size selective transport) - Does the "bed material" consider the bed surface or substrate? Please specify. - The bed material nicely shows downstream fining. Please explain. Is this due to preferential deposition of coarse sediment or particle abrasion? Or a combination?

12. I'd change the order of the research questions in Ln 94-96 "Is the entrainment formulation really necessary when modeling the LYR? Or more specifically, under what circumstances should a numerical modeler be impelled to implement the entrainment formulation instead of the flux formulation for river morphodynamic modeling?" to 1. "Under what conditions should one apply an entrainment form or flux form description of conservation of sediment mass?" 2. "Which form of the sediment conservation equation is most suitable for LYR?"

13. Ln 118. No bedrock. Please explain to the reader how valid this assumption is.

14. Ln 119. I think here you say that you impose a constant flow rate and sediment supply rate at the upstream end. What impact does this assumption have on the model results and conclusions? Please address this in the discussion section. Also holds for ln 133-137.

15. Ln 164 or Eq.7. It is about the interface elevation and not the bed surface elevation. The parameter La is missing. Right-hand terms should read d(zb-La)/dt<0 and d(zb-La)/dt>0.

16. Ln 226-230. The friction term in the original E&H formulation includes form drag. If there is barely any form drag such as in the LYR, then it makes sense that the original E&H does not do well and needs to be adjusted.

17. Ln 232-247. It may be interesting to mention that a fractional version of E&H was first proposed by Van der Scheer et al. (2002) and was later used by Blom et al. (2016, 2017a, 2017b).

18. Ln 310. Please be specific. "Slower degradation" refers to a slower downstream propagation of the degradational wave? Differences between Figs 3a and 4a are difficult to see anyway..

19. Ln 310. "more diffusive sediment load reduction". Or a faster downstream propagation of the disturbance?

20. Ln 342. 0.05. Why so extreme? Ln 483. 20. Why so extreme?

21. Ln 363-364. "The sediment supply rate of each grain size range is 364 set at 10% of its equilibrium sediment transport rate. This results in .... and a grain size distribution of the sediment supply ... that is identical to the grain size distribution of the equilibrium sediment load." Here it is essential for the reader to understand that the GSD of the sediment supply does not change: only the total sediment supply is reduced by 90%. This means that also the equilibrium GSD of the suspended load must be the same as the one of the sediment supply and so does not change with time. The equilibrium bed surface texture gets coarser with a reduction of the total sediment supply (Blom et al. 2016, 2017a). This is because with a reduced total sediment supply the equilibrium flow velocity decreases and the mobility difference between the grain size fractions increases. This implies that with a decrease of the total sediment supply the bed surface needs to coarsen to allow for the supplied sediment to be transported downstream. These things need to be explained to the reader.

22. Ln 372. "with at least two kinematic waves". Each grain size fraction induces the migration of a perturbation (Stecca et al., 2014, 2016). It would be nice to illustrate this.

23. Ln 376. See previous two comments. I think it would be illustrative and helpful if the authors would validate whether, if they continue their runs for a very long time, the

GSD of the suspended load becomes equal to the GSD of the sediment supply.

24. Ln 438. The authors neglect the temporal derivative. Can the authors quantify or justify this assumption?

25. Figs 6 and 7. Shouldn't the equilibrium channel slope and bed surface texture (Blom et al, 2017a) be equal for the flux form and the entrainment form? It would be nice if the authors would confirm this.

26. Section 3.2. Has the value of the active layer thickness been provided, and the total number of grain size fractions n? What is the height of the grid cells used to register the surface and substrate GSD? The flow is solved using a Godunov type scheme. What about the conservation equations for sediment mass? How are they solved?

27. The authors mention that they fix the downstream bed elevation by assuming normal flow at the downstream end. Yet, later the authors seem to mention that actually they put the downstream boundary condition sufficiently far to avoid backwater effects. Isn't this a contradiction?

28. In the simulation results one observes that changes in grain size arrive at the downstream end although bed elevation is constant with time. I do not understand this.

29. Has the CFL criterion for modelling bed elevation and the flow been considered?

30. Section 3.2. Have the authors experienced any unreasonable instabilities in their numerical runs (Chavarrias et al, 2018)?

31. Figure 9. The adaptation length is represented by 3-4 cells. Could the conclusion that the adaptation is irrelevant be due to not well solving it?

32. Section 4.2. I think the authors may like to consider these results in the context of the results of Stecca et al (2014, 2016).

33. Equation (37). The authors treat only one fraction and consider the equation to be an advection equation. In reality they have a system of advection equations in which

the source terms links them. This may yield different behavior.

34. Equation (38). The authors are not the first ones. I think the authors should compare these results to the ones of Stecca (2014, 2016).

35. Lines 494-498: I'd propose to rephrase.

36. Line 557-558. The authors say the entrainment form is needed when studying sorting processes. I do not think this conclusion can be drawn.

37. The conclusion section reads as a summary. I'd recommend revision and limiting the section to conclusions.

38. Ln 643-646. Is the entrainment form recommended provided that all 3 requirements are fulfilled or if only 1 of the 3 is fulfilled?

39. Ln 643-646. Please provide some more information here. When more information is added this finding should be one of the main results, I think. Also see suggestion on research questions.

40. Line 681. Please explain why the time derivatives may be omitted.

Minor comments:

41. Ln 17 and 45. alternate → alternative?

42. Ln 35. Reference to Parker 2004 can be omitted.

43. Ln 85. bed material → surface or substrate?

44. Ln 90. "and thus more likely to be" → "as it is"

45. Ln 141. I think the 0.4 value should be listed at a later point in the manuscript.

46. Ln 154. "La is often related to the height of dunes so that" → "La is often related to the height of dunes (Blom, 2008) so that"

47. Ln 161 and 162. You'll need to apply Eq (6) to n-1 sediment fractions.

48. Ln 248-249. I'd rephrase 'hiding effects between coarse and fine sediment'.

49. Ln 267. Blom et al 2003 → Blom 2008

50. Ln 277 and 621. I'd avoid using "=" like this in a sentence.

51. Ln 283 and 284. I'd change the unit years to something much smaller.

52. Ln 285-287. I'd rephrase the following sentence: "But it should be noted that the aim of this paper is not to reproduce specific aspects of the morphodynamic processes of LYR, but to compare the flux form and entrainment form of Exner equation in the context of conditions typical of LYR."

53. Fig 2. Caption and legend. "Initial bed" refers to surface or substrate? "Washload sizes" refers to which sizes?

54. Ln 436-437. Why repeat an equation?

55. Ln 621. I'd avoid starting a sentence with "But".

56. Ln 617 Sentence starting with "Moreover, . . . unchanged". I'd omit this.

References Armanini, A., Di Silvio, G., 1988. A one-dimensional model for the transport of a sediment mixture in non-equilibrium conditions. J. Hydraul. Res. 26 (3), 275–292. https://doi.org/10.1080/002216 88809499212. Bell, R.G., Sutherland, A.J., 1983. Nonequilibrium bedload transport by steady flows. J. Hydraul. Eng. 109 (3). https://doi.org/10.1061/(ASCE)0733-9429(1983)109:3(351). Blom, A., 2008. Different approaches to handling vertical and streamwise sorting in modeling river morphodynamics. Water Resour. Res. 44 (3), W03415. https://doi.org/10.1029/2006WR005474. Blom, A., Arkesteijn, L., Chavarrías, V., Viparelli, E., 2017. The equilibrium alluvial river under variable flow and its channel-forming discharge. J. Geophys. Res. Earth Surf., https://doi.org/10.1002/2017JF004213. Blom, A., Parker, G., 2004. Vertical sorting and the morphodynamics of bed-form dominated rivers: a modeling framework. J. Geophys. Res. Earth Surf. 109, F02007. https://doi.org/10.1029/2003JF000069.

[Figure]

Blom, A., Parker, G., Ribberink, J.S., de Vriend, H.J., 2006. Vertical sorting and the morphodynamics of bedform-dominated rivers: an equilibrium sorting model. J. Geophys. Res. Earth Surf. 111, F01006. https://doi.org/10.1029/2004JF000175. Blom, A., Ribberink, J.S., Parker, G., 2008. Vertical sorting and the morphodynamics of bed form-dominated rivers: a sorting evolution model. J. Geophys. Res. Earth Surf. 113, F01019. https://doi.org/10.1029/2006JF000618. Blom, A., Viparelli, E., Chavarrías, V., 2016. The graded alluvial river: profile concavity and downstream fining. Geophys. Res. Lett. 43, 1–9. https://doi.org/10.1002/2016GL068898. Chavarrías, V., Stecca, G., and A. Blom, 2018. Ill-posedness in modeling mixed sediment river morphodynamics. Adv. Water Resour., doi:10.1016/j.advwatres.2018.02.011 Stecca, G., Siviglia, A., Blom, A., 2014. Mathematical analysis of the Saint-Venant-Hirano model for mixed-sediment morphodynamics. Water Resour. Res. 50, 7563–7589. https://doi.org/10.1002/2014WR015251. Stecca, G., Siviglia, A., Blom, A., 2016. An accurate numerical solution to the Saint- Venant-Hirano model for mixed-sediment morphodynamics in rivers. Adv. Water Resour. 93, Part A, 39–61. https://doi.org/10.1016/j.advwatres.2015.05.022. Van der Scheer, P., J.S. Ribberink, and A. Blom (2002), Transport formulas for graded sediment; Behaviour of transport formulas and verification with data. Research Report 2002R-002, Civil Engineering, University of Twente, Netherlands

---

## Referee Comment (RC2) · Anonymous Referee #2 · 17 Jul 2018

I have read the comments of Anonymous Referee #1, and I am in agreement with the general comments and most of the specific comments in that review. I have not repeated detailed comments here, except where to add further points to that referee's analysis.

The paper presents a very detailed and clear explanation of two related, but different, approaches to modelling suspended sediment transport in a large lowland river. The description and explanation of the model and the key equations is very helpful, and could itself become an excellent reference source in future. The substantive results of the paper are interesting, although somewhat unsurprising. Referee #1 notes that

the main question in the paper is actually "When does a modeller need to properly account for the lag of suspended load transport?". I concur with this point, and also that the flux form of the conservation equation is not necessarily synonymous with a capacity-based model.

Main comments:

1. My major concern is the generality of the results given the effect of the upstream boundary on model results in all cases where the upstream sediment input is less than capacity. In these cases, there is degradation at the upstream boundary and a gradual, exponential, rise towards capacity transport at some distance downstream. Such a situation is appropriate immediately below a dam, but without such a barrier to sediment transport the transport rate (sediment input) at the boundary would be in equilibrium with local flow conditions and availability of sediment on the river bed.

2. Much of the paper addresses the conditions under which the model predicts diffusion and/or advection. The analysis of this aspect is very useful and provides a way of evaluating how perturbations should be translated downstream. There is no consideration of the extent to which the model behaviour may reflect numerical behaviour. Morphological models of this type tend to be diffusive, for reasons that are considered in the paper and which relate to the damping effect as the bed surface (gradient and grain size) co-evolve in response to divergence in the sediment transport rate. Advection is not surprising where significant disturbances are introduced, especially where transport rates are relatively high. However, I find it difficult to understand how the model can generate successive advective waves (lines 627-35 are convincing, but for one wave), and wonder if there are aspects to the behaviour of the sediment transport function and/or sediment routing that lead to this. For example, it is unclear if the active layer thickness is maintained (ie is the surface layer mixed with the sub-surface at every timestep, or is the active layer exhausted and then replenished from the subsurface only after this exhaustion is complete? This latter approach could readily lead to generation of additional waves of adjustment as the sub-surface will be finer than

the surface in the surface layer).

3. Lines 248-9: following the previous comment, it would be useful to know a little more about the way that equations (24)-(27) work. Did Ma et al (2017) use a size-specific formulation too?

4. Lines 88-90: I am unclear that the entrainment-based approach is more physically based than the flux-based approach. A properly-calibrated transport model should work just as well for the flux-based approach – if this predicts transport rates that exceed observations, this suggests to me a problem with the transport equation rather than needing a 'supply-limitation' correction factor applying. I also do not see the relevance of saying that Chinese researchers have a particular approach.

5. Line 92: is the additional computational requirement significant – I suspect not.

6. Lines 117-8 (and others): using a simplified geometry is entirely justified and makes a lot of sense. However, some consideration of the potential significance of these assumptions would be useful. For example, downstream of the dam is degradation uniform across the channel or is it concentrated in a thalweg leading to asymmetric cross-section geometry? On lines 253-4, can some indication be given about the observed variation in width and slope? I assume that there are no significant tributary inflows of water and/or sediment, and this could be stated here too.

7. Following the previous comment, it would be good to have some more assessment of model performance. Some annual flux estimate comparisons are made, which are encouraging. Are there other pieces of evidence (eg order of magnitude of degradation below the dam, and distance of propagation of the degradational wave after some years) that can be used to provide a general validation of the model?

8. Line 156 (and others): given the seasonal flow regime of this river, there will be variation in dune height during the sediment transporting period. I appreciate the use of constant flow and La for these simulations, but could the effect of La changing in

time be considered?

9. Model comparison (Line 364) – using absolute values is actually less informative than retaining the signs (or using yE/yF ratios). Table 2 (and later tables) – I did not find these tables very helpful, and wonder if it would be better to put this information onto the relevant figures as an additional plot (Lines 414-422 are very wordy as a result of describing Table 3 – would be easier to describe a graph of the same results). The figures on the table definitely do not need to be 2 decimal places.

10. Line 344: 'intentionally unrealistic' is ok, but readers might assume linear behaviour between your realistic and unrealistic boundary conditions. Do you know if this is a reasonable assumption to make?

11. Lines 451-2: I think (from memory – don't have the paper to hand) that this is similar to Philipps and Sutherland's formulation. If so, maybe reference this here.

12. Lines 467-8: another way to interpret this is in terms of settling velocity (ie there is a settling velocity, or Stokes number, above which adaptation length can be ignored).

13. Lines 555-7: I am not sure that the results in this paper can be used to make recommendations about event-scale modelling. It would be good to see some event-scale simulations to assess the significance of any differences between the two formulations.

Minor comments:

1. Line 28: I would say 'adapted for' rather than 'designed for'

2. Line 36: 'some aspect of sediment transport' is rather imprecise – can this be made more specific.

3. Line 119: presumably the grain-size distribution of the flux is also fixed?

4. Lines 271-2: can the sorting of the gsd also be given (maybe use the blank spaces on Figure 2 to put these numbers onto)?

5. Line 281: I think it is 400 cells, but 401 computational nodes.

6. Line 282: can you give Courant numbers for these timesteps?

7. Table 1: Add dx and dt to this – the table is a very useful quick reference point, so having these values here would be informative.

8. Figure 3: Could the water surface be added to this plot (maybe initial and final values only)?

---

## Author Comment (AC1) · 19 Aug 2018

Please see the attached zip file for our author comments. The zip file includes four pdf files.

(1) Response to referees' comments; (2) New main text; (3) Main text with track changes; (4) New supplement.

Please also note the supplement to this comment:
https://www.earth-surf-dynam-discuss.net/esurf-2018-42/esurf-2018-42-AC1-supplement.zip

[Figure]

**ESurfD**

Interactive
comment

---

## Author Response (AR1)

**Response file to comments**

**Reviewer #1**

1. It seems that the main question the authors are answering ("what are the difference between an entrainment form conservation equation and when to use a flux-based formulation") rather is "When does a modeler need to properly account for suspended load mechanisms?"

We have rephrased the questions, and their order in the new manuscript. They now more closely reflect the reviewer's comments. See Lines 100-105 of the manuscript with track changes.

2. The authors treat the flux form of the mass conservation equation as a synonym to a capacity-based or equilibrium approach, and the entrainment form of the mass conservation equation is considered noncapacity-based or non-equilibrium. I think the two types of models (flux form and entrainment form) are not synonyms for capacity-based and noncapacity-based. These terms are not equivalent. Revising this may impact several parts of the text.

We have now specifically defined what we mean by the flux form and entrainment form. See Lines 23-24 of the manuscript with track changes. We define again here: "Here we identify the flux form as based on the local capacity sediment transport rate, and the entrainment form as based on the local capacity entrainment rate". Although we do not go into details in the paper, we note here that our system (the Yellow River) is suspension-dominated to the point where bedload is negligible. The phenomenon documented by Bell and Sutherland (1983) can be quantified using an entrainment-based *bedload* formulation such as Pelosi and Parker (2014). We note here that El kadi Abderrezzak and Pacquier (2009) list the form $dQ_s/dx = (Q_s^{cap} - Q_s)/L_{ad}$. where $L_{ad}$ is the adaptation length. This is nothing more and nothing less than an entrainment form, but one for which the guts of the adaptation length $L_{ad}$ remain undefined. In our system $L_{ad}$ is specifically given as $q_w/(v_s r_0)$, where $q_w$ is the water discharge per unit width and $v_s$ is fall velocity.

3. Associated with the previous comment: Could Bell and Sutherland (1983) and Armanini and Di Silvio (1988) be examples of the flux-form combined with a non-capacity approach?

Bell and Sutherland (1993) is by and large a bedload entrainment formulation with an extra term. See their Eq. (14) therein. The term $g_{se}$ denotes the locally-computed capacity transport rate. Armanini and Di Silvio (1988) is complex and hard to interpret.

4. Associated with the same comment: And may Blom and Parker (2004), Blom et al. (2006, 2008) be an example of the entrainment form combined with a capacity-based approach?

We are happy to reply, "yes!" They have the distinguishing feature of, you guessed it, eliminating the need for an active layer!

5. Isn't the Exner equation of conservation of sediment mass a flux-based approach? Shouldn't the authors consider another (other than "Exner") name for the entrainment form of conservation of sediment mass?

The flux form of Exner is:

$$\left(1-\lambda_p\right)\frac{\partial \eta}{\partial t} = -\frac{\partial q_s}{\partial x}$$

The entrainment form is:

$$\left(1-\lambda_p\right)\frac{\partial \eta}{\partial t} = -v_s\left(E-r_oC\right)$$

$$\frac{\partial Ch}{\partial t} + \frac{\partial q_s}{\partial x} = v_s\left(E-r_oC\right)$$

The first and second equations above are alternative forms for sediment mass conservation (Exner). We have not modified the paper in this regard. The third equation can be written as,

$$\frac{\partial Ch}{\partial t} + \frac{\partial q_s}{\partial x} = \frac{1}{L_{ad}}\left(q_{se}-q_s\right) \quad , \quad q_{se} = \frac{E}{r_o}q_w \quad , \quad L_{ad} = \frac{q_w}{r_o v_s}$$

Thus it fits within the generic form below, which can be found in Bell and Sutherland (1983) and El kadi Abderrezzak and Paquier (2009).

$$???+\frac{\partial q_s}{\partial x} = \frac{1}{L_{adapt}}\left(q_{se}-q_s\right)$$

We have not modified the text: both papers are already quoted in the text.

6. Ln 26 "study this problem by comparing the results of flux-based and entrainment-based morphodynamics". Please explain in what way you are comparing the two types of conservation models. This also holds for ln 105-106. One of the ways seems to be by mimicking "the reduction of the sediment load in the LYR in recent years" (Ln 279). I think this information needs to be moved to the Introduction section. Figs 6 and 7. Shouldn't the equilibrium channel slope and bed surface texture be equal for the flux form and the entrainment form? It would be nice if the authors would confirm this.

The reviewer is right that the comparison is conducted based on the premise of a reduction of sediment supply, so as to mimic the effect of the operation of Xiaolangdi Dam. We have explained this in the manuscript. See Line 30 and Lines 108-109 of the manuscript with track changes.

In terms of the channel equilibrium predicted by the flux and the entrainment forms, actually we have explained this in Section 4.3 in the original manuscript. See Lines 609-611 of the manuscript with track changes.

7. Ln 26 "study this problem by comparing the results of flux-based and entrainment-based morphodynamics under conditions typical of the Lower Yellow River". I would suggest to develop generic guidelines for when to use an entrainment form conservation equation and when to use a flux-based formulation, and only after this move toward the case of the Lower Yellow River.

While we have not performed a general study, we have performed the calculations for the evaluation requested by the reviewer. In Line 522 of the manuscript with track changes, we point out that the adaptation length $L_{ad}$ for uniform material is only 1.88 km, a value that is small compared to the reach length of 200 km. We identify this as the reason that the flux formulation can be used; i.e. $L_{ad}/L \ll 1$. We also show that this condition does not holds for the finer sizes typical bed material of the Yellow River. We have modified a sentence to read "In this case and in general, the predictions of the flux form and the entrainment form show little difference when $L_{ad}/L \ll 1$, where $L$ is domain length." See Lines 523-524 of the manuscript with track changes.

8. Ln 45-47. I'd rephrase this as application of the entrainment form of the conservation equation is not necessarily limited to suspended sediment.

The sentence has been rephrased so that the entrainment form is not limited to suspended sediment. See Lines 50-51 of the manuscript with track changes.

9. Ln 57-71. These lag issues are not only covered by the entrainment form, but may also be covered by the flux form... Also see comment 2.

We do not think so. As we have described in response to Query 2, we believe that *all* formulations of the form $dQ_s/dx = (Q_s^{cap} - Q_s)/L_{ad}$ are entrainment forms, in which the entrainment rate $\sim Q_s^{cap}/q_w$. Again, this holds regardless of whether the mode of transport is bedload or suspended load.

10. Ln 78. "More recently, however, since the operation of Xiaolangdi Dam in 1999 the LYR has seen a substantial reduction in its sediment load (Fig. 1(b))". - The authors do not address the distinct temporal decrease of the annual sediment load between 1950-2000. - Why is the year 2000 indicated in Figure 1b and not 1999? Please indicate in the figure whether 2000 is supposed to indicate the year of the Xiaolangdi Dam construction.

The reviewer is right that there is a typo in Figure 1(b). The year 1999, rather than 2000, should be used which indicates the time of the operation of the Xiaolangdi Dam. The figure has been replotted as suggested by the reviewer. We have also rephrased the sentence so that the decrease of sediment load in recent decades is addressed. See Line 82 of the manuscript with track changes.

11. Figure 1 shows a number of very interesting features that are currently not addressed by the authors: - the distinct temporal decrease of the annual sediment load between 1950-2000. What is the cause of this decrease? - The three lines for the three cities are highly correlated. Please indicate and explain. - The suspended load is significantly finer than the bed material (i.e., grain size selective transport) – Does the "bed material" consider the bed surface or substrate? Please specify. - The bed material nicely shows downstream fining. Please explain. Is this due to preferential deposition of coarse sediment or particle abrasion? Or a combination?

(1) The decrease in sediment load prior to 2000 is a basinwide phenomenon that is not directly germane to this paper. We can understand why the issues caught the eye of the reviewer. We consulted our co-author, Yuanfeng Zhang, who is perhaps the foremost (in China and thus the world) expert on sediment transport in the Yellow River. This is what he said. "There are 3 main reasons for the sediment load decrease from 1970 (strictly say is that from 1980s for the LYR ) to 2000: (a) revegetation, (b) terrace fields, (c) checkdam and small dam (we saw lots of them in 2017), and also other reasons such as taking sediment from river for construction, climate changes etc." We have not added this to the paper, as we do not wish to focus on basinwide sediment production.

(2) Of the three lines for load, note that there is a consistent decline from Huayuankou (upstream) to Lijin (downstream). This is consistent with a river in depositional mode in the giant alluvial fan of the North China Plain.

(3) We are not aware of any studies of the abrasion of silt in water. The particles are likely too fine to undergo comminution due to collision.

12. I'd change the order of the research questions in Ln 94-96 "Is the entrainment formulation really necessary when modeling the LYR? Or more specifically, under what circumstances should a numerical modeler be impelled to implement the entrainment formulation instead of the flux formulation for river morphodynamic modeling?" to 1. "Under what conditions should one apply an entrainment form or flux form description of conservation of sediment mass?" 2. "Which form of the sediment conservation equation is most suitable for LYR?"

The research questions have been rephrased according to the suggestion of the reviewer. See Lines 100-105 of the manuscript with track changes.

13. Ln 118. No bedrock. Please explain to the reader how valid this assumption is.

We again consulted Yuanfeng Zhang about this. "The LYR has been heavy sediment-laden and depositional river for hundreds or thousands years. So, definitely, no exposed bedrock in the LYR". We have modified the text. See Lines 131-132 of the manuscript with track changes.

14. Ln 119. I think here you say that you impose a constant flow rate and sediment supply rate at the upstream end. What impact does this assumption have on the model results and conclusions? Please address this in the discussion section. Also holds for Lines 133-137.

We present numerical cases with hydrographs in the new Supplement. Results indicate that our conclusions based on constant flow discharge also hold when hydrographs are implemented. See Section S2 and S3 of the Supplement.

We have also moved Appendix A and Appendix B to the Supplement.

15. Ln 164 or Eq.7. It is about the interface elevation and not the bed surface elevation. The parameter La is missing. Right-hand terms should read d(zb-La)/dt<0 and d(zb-La)/dt>0.

The reviewer is right that $d(z_b-L_a)$, rather than $dz_b$, should be used in the right-hand terms. But since a constant active layer thickness $L_a$ is implemented in this paper, the two forms are identical. We have revised the equation as suggested by the reviewer, so that it is more universal.

16. Ln 226-230. The friction term in the original E&H formulation includes form drag. If there is barely any form drag such as in the LYR, then it makes sense that the original E&H does not do well and needs to be adjusted.

We agree with the reviewer. Actually we think that the lack of form drag in the LYR leading to very high sediment transport rates, which cannot be reproduced by the original E&H formulation. We quote Ma et al. (2017) in this regard.

17. Ln 232-247. It may be interesting to mention that a fractional version of E&H was first proposed by Van der Scheer et al. (2002) and was later used by Blom et al. (2016, 2017a, 2017b).

We have added the references. See Lines 274-276 of the manuscript with track changes.

18. Ln 310. Please be specific. "Slower degradation" refers to a slower downstream propagation of the degradational wave? Differences between Figs 3a and 4a are difficult to see anyway.

We have added insets in Figure 3-8 to show the detailed results of the bed elevation near the upstream end. A comparison between Figure 3(a) and Figure 4(a) shows that the flux form predicts a 3 m degradation at the upstream end whereas the entrainment form predicts a 2.3 m degradation. This is the reason why we say "slower degradation". See Lines 341-342 of the manuscript with track changes.

19. Ln 310. "more diffusive sediment load reduction". Or a faster downstream propagation of the disturbance?

We think that they are the same thing.

20. Ln 342. 0.05. Why so extreme? Ln 483. 20. Why so extreme?

These intentionally extreme values allow us to explore limiting behavior. The reason for choosing extreme values is to understand under what conditions the flux form and the entrainment form lead to different/similar predictions, as explained in the manuscript. We have specifically demonstrated that such values are unrealistic. See Line 381 and Line 461 of the manuscript with track changes.

21. Ln 363-364. "The sediment supply rate of each grain size range is set at 10% of its equilibrium sediment transport rate. This results in… and a grain size distribution of the sediment supply… that is identical to the grain size distribution of the equilibrium sediment load." Here it is essential for the reader to understand that the GSD of the sediment supply does not change: only the total sediment supply is reduced by 90%. This means that also the equilibrium GSD of the suspended load must be the same as the one of the sediment supply and so does not change with time. The equilibrium bed surface texture gets coarser with a reduction of the total sediment supply (Blom et al. 2016, 2017a). This is because with a reduced total sediment supply the equilibrium flow velocity decreases and the mobility difference between the grain size fractions increases. This implies that with a decrease of the total sediment supply the bed surface needs to coarsen to allow for the supplied sediment to be transported downstream. These things need to be explained to the reader.

We have explained in the manuscript that the GSD of the sediment supply does not change and only the total sediment supply is reduced by 90%. See Lines 411-412 of the manuscript with track changes. As for the surface coarsening, we have added the following line to the text. "This represents armoring, mediated by the hiding functions of Eqs. (26) and (27)." See Lines 423-424 of the manuscript with track changes.

22. Ln 372. "with at least two kinematic waves". Each grain size fraction induces the migration of a perturbation (Stecca et al., 2014, 2016). It would be nice to illustrate this.

The two reference should have been in the original text. They have now been added to the manuscript. See Lines 418-419 of the manuscript with track changes. By way of apology, we mention that Stecca (2014) is referenced copiously in our recent paper below.
An, C-G., Fu, X-D., Wang, G-Q and Parker, G. 2017. Effect of Grain Sorting on Gravel-bed River Evolution Subject to Cycled Hydrograph: Bedload Sheets and Breakdown of the Hydrograph Boundary Layer. Journal of Geophysical Research, 122(8), 1513-1533, DOI: 10.1002/2016JF003994.

23. Ln 376. See previous two comments. I think it would be illustrative and helpful if the authors would validate whether, if they continue their runs for a very long time, the GSD of the suspended load becomes equal to the GSD of the sediment supply.

In Section 4.3 we have demonstrated that under equilibrium condition, the sediment transport rate of each grain size equals to the sediment supply rate of each grain size. This statement indicates that the GSD of suspended load equals the GSD of sediment supply. See Lines 609-611 of the manuscript with track changes.

24. Ln 438. The authors neglect the temporal derivative. Can the authors quantify or justify this assumption?

We neglect the temporal derivative with the purpose to simplify the mathematical analysis and characterize the adaptation length scale (rather than the adaptation time scale). The adaptation time scale can be obtained if we neglect the spatial derivative and conduct a similar analysis. We explain this in the manuscript. See Line 498 of the manuscript with track changes.

25. Figs 6 and 7. Shouldn't the equilibrium channel slope and bed surface texture (Blom et al, 2017a) be equal for the flux form and the entrainment form? It would be nice if the authors would confirm this.

Yes, we have confirmed this in Section 4.3. See Lines 609-611 of the manuscript with track changes.

26. Section 3.2. Has the value of the active layer thickness been provided, and the total number of grain size fractions n? What is the height of the grid cells used to register the surface and substrate GSD? The flow is solved using a Godunov type scheme. What about the conservation equations for sediment mass? How are they solved?

The value of active layer thickness ($L_a$ = 0.738) was provided at the beginning of Section 3. See Line 296 of the manuscript with track changes. The total number of grain size fractions can be inferred from Figure 2: $n$ = 5. We have added this information to the text. See Line 305 of the manuscript with track changes. The height of grid cell to register the surface GSD is $L_a$. The height to register the substrate GSD is 0.5 m. This information has been added to the text. See Line 297 of the manuscript with track changes. For the conservation equations for sediment mass, we implemented a first-order explicit scheme for the temporal derivatives and a first-order upwinded scheme for the spatial derivatives.

27. The authors mention that they fix the downstream bed elevation by assuming normal flow at the downstream end. Yet, later the authors seem to mention that actually they put the downstream boundary condition sufficiently far to avoid backwater effects. Isn't this a contradiction?

If we put the downstream boundary condition close to the river mouth, the backwater effects would be so strong that the normal flow assumption would not be appropriate. Therefore, we put the downstream boundary sufficiently far upstream of the mouth so that the normal flow can be implemented as the downstream boundary condition.

28. In the simulation results one observes that changes in grain size arrive at the downstream end although bed elevation is constant with time. I do not understand this.
HERE

The active layer is taken to be 0.738 m in thickness. It adjusts much more quickly than bed elevation itself. You will find the same result in:
An, C-G., Fu, X-D., Wang, G-Q and Parker, G. 2017. Effect of Grain Sorting on Gravel-bed River Evolution Subject to Cycled Hydrograph: Bedload Sheets and Breakdown of the Hydrograph Boundary Layer. Journal of Geophysical Research, 122(8), 1513-1533, DOI: 10.1002/2016JF003994.

29. Has the CFL criterion for modelling bed elevation and the flow been considered?

The CFL criterion has been considered such that that the time steps implemented in the simulation are small enough to avoid numerical instabilities.

30. Section 3.2. Have the authors experienced any unreasonable instabilities in their numerical runs (Chavarrias et al, 2018)?

Even though Chavarrias et al. (2018) has reported that unreasonable instabilities could occur under certain circumstances when modeling mixed sediment river morphodynamics, we did not run into such problems in our simulations. We have mentioned this in the manuscript. See Lines 673-675 of the manuscript with track changes.

31. Figure 9. The adaptation length is represented by 3-4 cells. Could the conclusion that the adaptation is irrelevant be due to not well solving it?

In order to check whether the adaptation is irrelevant be due to poorly solving for it, we recalculated the case of uniform sediment (in Section 3.1) with a smaller cell size. The cell size specified here is $\Delta x = 250$ m, which is half the cell size specified in the manuscript. Figure R1 and Figure R2 show the modeling results using the flux form and the entrainment form of the Exner equation, respectively.

If we compare Figure R1 with Figure 3 in the MS, and Figure R2 with Figure 4 in the MS, we can see that reducing the cell size from 500 m to 250 m has almost no influence on the simulation results, in terms of both the flux form and the entrainment form of the Exner equation. So we think that our solution might be pretty good!

[Figure]

Figure R1. Case of uniform sediment using the flux form of Exner equation and a cell size of 250 m (half of that implemented in the MS). Time variation of (a) bed elevation $z_b$ and (b) sediment load per unit width $q_s$ in response to the cutoff of sediment supply.

[Figure]

Figure R2. Case of uniform sediment using the entrainment form of Exner equation and a cell size of 250 m (half of that implemented in the MS). Time variation of (a) bed elevation $z_b$ and (b) sediment load per unit width $q_s$ in response to the cutoff of sediment supply.

32. Section 4.2. I think the authors may like to consider these results in the context of the results of Stecca et al (2014, 2016).

The papers of Stecca et al. (2014, 2016) should have been referenced in the original manuscript. They are now referenced in the new manuscript. See Lines 560-562 of the manuscript with track changes.

33. Equation (37). The authors treat only one fraction and consider the equation to be an advection equation. In reality they have a system of advection equations in which the source terms links them. This may yield different behavior.

We think that in Section 4.2 we actually treat the equation system rather than only one equation, because Eq. (37) holds for each size fraction. If we list $i$ from 1 to $n$-1, we get the equation system.

But here we only show the general formulation of the equation system, which is enough to explain our modeling results. As for the interaction among different size fractions, it is represented in the source term of Eq. (39) via the terms with subscription $j$ ($j \neq i$).

34. Equation (38). The authors are not the first ones. I think the authors should compare these results to the ones of Stecca (2014, 2016).

The reviewer is right that we are not the first ones to study this problem. The mathematical analysis of Stecca et al. (2014, 2016) is a bit different form our analysis in that they implemented the St. Venant equation as well as a linearized analysis which is only applicable for small perturbations. But the overall characteristics of the governing equations derived by Stecca et al. (2014, 2016) and this paper are similar. Since the purpose of the mathematical analysis in this section is to explain the modeling results in Section 3, we do not think that it is necessary to compare our analysis with that of Stecca et al. (2014, 2016) in detail at this point in the paper. We have, however, referenced Stecca et al. (2014, 2016) and related papers in the new manuscript. See Lines 560-562 of the manuscript with track changes.

35. Lines 494-498: I'd propose to rephrase.

We have slightly reworded the text. See Lines 555-560 of the manuscript with track changes.

36. Line 557-558. The authors say the entrainment form is needed when studying sorting processes. I do not think this conclusion can be drawn.

We appreciate the reviewer for the correction. Actually what we want to say is that the entrainment form is needed when studying the sorting processes of fine-grained sediment. This is because fine-grained sediment has a large adaptation length $L_a$ and a large diffusivity coefficient $\nu_i$. The flux form can lead to an overestimation of advection as it pertains to sorting processes under such circumstances. We have revised the text to make our statement more clear. See Lines 618-634 of the manuscript with track changes.

37. The conclusion section reads as a summary. I'd recommend revision and limiting the section to conclusions.

We have rephrased the Conclusion a bit. Generally we prefer the Conclusion the way it is.

38. Ln 643-646. Is the entrainment form recommended provided that all 3 requirements are fulfilled or if only 1 of the 3 is fulfilled?

We have revised the text to make our conclusion clear. In the new text we do not specifically enumerate the requirements, but yes, we recommend the entrainment formulation when all are fulfilled. See Lines 618-634 of the manuscript with track changes.

39. Ln 643-646. Please provide some more information here. When more information is added this finding should be one of the main results, I think. Also see suggestion on research questions.

We have expanded this in the new text. See Lines 726-732 of the manuscript with track changes.

40. Line 681. Please explain why the time derivatives may be omitted.

In principle, the time derivatives cannot be omitted. But for the purpose of illustrating the spatial lag, the analysis is clear if we omit the time derivatives. By the same token, in order to illustrate temporal lag, it is clearer to omit the spatial derivatives.

41. Ln 17 and 45. alternate → alternative?

The manuscript has been revised as suggested by the reviewer.

42. Ln 35. Reference to Parker 2004 can be omitted.

Reference to Parker 2004 has been deleted here.

43. Ln 85. bed material → surface or substrate?

The data in Figure 1(c) is based on the sampling of the bed surface. We have modified the text to clarify this. See Line 91 of the manuscript with track changes.

44. Ln 90. "and thus more likely to be" → "as it is".

The manuscript has been revised as suggested by the reviewer.

45. Ln 141. I think the 0.4 value should be listed at a later point in the manuscript.

The 0.4 value has been moved to Section 3. See Line 305 of the manuscript with track changes.

46. Ln 154. "La is often related to the height of dunes so that" → "La is often related to the height of dunes (Blom, 2008) so that".

The manuscript has been revised as suggested by the reviewer.

47. Ln 161 and 162. You'll need to apply Eq (6) to n-1 sediment fractions.

The reviewer is right that Eq (6) is applied to n-1 sediment fractions in the model. Here we only show the general formulation of the equation system.

48. Ln 248-249. I'd rephrase 'hiding effects between coarse and fine sediment'.

The manuscript has been rewritten to explain the way that equations (24)-(27) work. See Lines

267-273 of the manuscript with track changes.

49. Ln 267. Blom et al 2003 → Blom 2008.

The reference has been changed as suggested by the reviewer.

50. Ln 277 and 621. I'd avoid using "=" like this in a sentence.

The "=" has been removed.

51. Ln 283 and 284. I'd change the unit years to something much smaller.

We would like to keep the unit years partly because the concept of flood intermittency factor is basically considered in the time scale of years.

52. Ln 285-287. I'd rephrase the following sentence: "But it should be noted that the aim of this paper is not to reproduce specific aspects of the morphodynamic processes of LYR, but to compare the flux form and entrainment form of Exner equation in the context of conditions typical of LYR."

We are not sure how to rephrase. The sentence correctly highlights the goals of the paper.

53. Fig 2. Caption and legend. "Initial bed" refers to surface or substrate? "Washload sizes" refers to which sizes?

"Initial bed" refers to both the initial surface and initial substrate. "Washload sizes" refer to grain sizes which belongs to the range of washload, i.e. < 15 μm.

54. Ln 436-437. Why repeat an equation?

We repeat for clarity. We do not want the reader to have to look many pages back to understand our argument.

55. Ln 621. I'd avoid starting a sentence with "But".

OK, changed.

56. Ln 617 Sentence starting with "Moreover, …unchanged". I'd omit this.

Three fairly large chunks of text related to arbitrarily increased or decreased fall velocity have been deleted from the Conclusion. See Lines 700-702, Lines 708-709, and Lines 714-718 of the manuscript with track changes.

**Reviewer #2**

1. My major concern is the generality of the results given the effect of the upstream boundary on model results in all cases where the upstream sediment input is less than capacity. In these cases, there is degradation at the upstream boundary and a gradual, exponential, rise towards capacity transport at some distance downstream. Such a situation is appropriate immediately below a dam, but without such a barrier to sediment transport the transport rate (sediment input) at the boundary would be in equilibrium with local flow conditions and availability of sediment on the river bed.

Morphodynamics is the essence of response to disequilibrium conditions inducing aggradation, degradation and sorting. Cutoff of sediment supply due to a dam is both an excellent way to study response to disequilibrium and characterize a key factor presently affecting the LYR.

In our simulations, we implement sediment supply rates which are less than sediment transport capacity. Such a situation is appropriate for the LYR or other alluvial rivers which are affected by dams. However, we think that the general conclusions of this paper are also applicable to other kind of perturbations which lead to an imbalance between the sediment supply and sediment transport capacity. Bed aggradation or degradation would occur from the place where perturbation is introduced, and sediment concentration (or sediment transport rate) would adjust exponentially to its equilibrium value.

As for the situation under which the sediment supply equals sediment transport capacity as suggested by the reviewer, we think that this situation does not shed light on our problem, because there would be no bed evolution, i.e. nothing would happen.

2. Much of the paper addresses the conditions under which the model predicts diffusion and/or advection. The analysis of this aspect is very useful and provides a way of evaluating how perturbations should be translated downstream. There is no consideration of the extent to which the model behavior may reflect numerical behavior. Morphological models of this type tend to be diffusive, for reasons that are considered in the paper and which relate to the damping effect as the bed surface (gradient and grain size) co-evolve in response to divergence in the sediment transport rate. Advection is not surprising where significant disturbances are introduced, especially where transport rates are relatively high. However, I find it difficult to understand how the model can generate successive advective waves (lines 627-35 are convincing, but for one wave), and wonder if there are aspects to the behaviour of the sediment transport function and/or sediment routing that lead to this. For example, it is unclear if the active layer thickness is maintained (ie is the surface layer mixed with the sub-surface at every timestep, or is the active layer exhausted and then replenished from the subsurface only after this exhaustion is complete? This latter approach could readily lead to generation of additional waves of adjustment as the sub-surface will be finer than the surface in the surface layer).

The issue of successive advective waves has been discussed thoroughly in Stecca et al. (2014, 2016). The number of kinematic waves actually depend on the discretization of the grain size distribution: each grain size fraction would induce the migration of one wave. We have explained this in the manuscript. See Lines 418-419 of the manuscript with track changes. The reviewer may also refer to Stecca et al. (2014, 2016) for more information.

When solving Equation (6) or Equation (15), the active layer exchange with the sediment suspended load as well as the substrate at the same time in every time step.

3. Lines 248-9: following the previous comment, it would be useful to know a little more about the way that equations (24)-(27) work. Did Ma et al (2017) use a size-specific formulation too?

The manuscript has been rewritten to explain the way that equations (24)-(27) work. See Lines 267-273 of the manuscript with track changes. Ma et al. (2017) did not use a size-specific formulation. Their relation calculates the total sediment transport rate with a characteristic grain size.

4. Lines 88-90: I am unclear that the entrainment-based approach is more physically based than the flux-based approach. A properly-calibrated transport model should work just as well for the flux-based approach – if this predicts transport rates that exceed observations, this suggests to me a problem with the transport equation rather than needing a 'supply-limitation' correction factor applying. I also do not see the relevance of saying that Chinese researchers have a particular approach.

We think that the entrainment-based approach is more physically based than the flux-based approach, because the entrainment-based approach considers lag effects (nonequilibrium sediment transport) whereas the flux-based approach cannot. The difference between the two approaches becomes more evident for finer sediment. We show that the difference is physically based. Solving the problem through calibration is not as satisfying as solving it through physics. We do indeed think that it is relevant to mention the Chinese LYR modeling approach heretofore, because we wish to influence this audience, as well as the general morphodynamics community.

5. Line 92: is the additional computational requirement significant – I suspect not.

According to our experience, the additional computational requirement is significant. The main reason is that the governing equations for sediment concentration need to be solved in the entrainment-based approach but do not need to be solved in the flux-based approach. For a sediment GSD with n size ranges, n more equations need to be solved every time step. Moreover, when dealing with fine sediment, a small time step is be needed to solve the additional governing equations of sediment concentration.

6. Lines 117-8 (and others): using a simplified geometry is entirely justified and makes a lot of sense. However, some consideration of the potential significance of these assumptions would be useful. For example, downstream of the dam is degradation uniform across the channel or is it concentrated in a thalweg leading to asymmetric cross-section geometry? On lines 253-4, can some indication be given about the observed variation in width and slope? I assume that there are no significant tributary inflows of water and/or sediment, and this could be stated here too.

Some explanation has been added to the manuscript as suggested by the reviewer. See Lines 127-137, and Lines 666-673 of the manuscript with track changes.

7. Following the previous comment, it would be good to have some more assessment of model performance. Some annual flux estimate comparisons are made, which are encouraging. Are there other pieces of evidence (eg order of magnitude of degradation below the dam, and distance of propagation of the degradational wave after some years) that can be used to provide a general validation of the model?

Our model overpredicts degradation. There are two reasons for this. 1) We have assumed that the sediment supply to the LYR is 10% of the pre-Xiaolangdi value. The actual number is closer to 20%. We chose the lower number in order to clearly see the difference between the two (flux and entrainment) models. In addition, our sediment transport relation does not undergo a phase change to lower sediment transport regime as the bed coarsens beyond 100 μm. We are working on a new sediment transport relation which includes this phase change. The issue is important, but not relevant to the major point of this paper, i.e. flux versus entrainment.

8. Line 156 (and others): given the seasonal flow regime of this river, there will be variation in dune height during the sediment transporting period. I appreciate the use of constant flow and La for these simulations, but could the effect of La changing in time be considered?

In this paper we relate the active layer thickness to the height of bedforms, which is 20% of the flow depth for the LYR according to Ma et al. (2017). A constant active layer thickness is implemented based on the equilibrium flow depth before the cutoff of sediment supply.

We agree with the reviewer that the dune height might change during the sediment transport period. But since the adjustment of bedforms is mostly much slower than the adjustment of flow hydraulics (i.e., the morphodynamic timescale is much larger than the hydraulic timescale), relating the active layer thickness directly to the instantaneous flow depth might be incorrect. Therefore, the assumption of constant active layer thickness can be regarded as an average for long-term hydraulic conditions. The question of how to quantify the instantaneous active layer thickness merits future study.

9. Model comparison (Line 364) – using absolute values is actually less informative than retaining the signs (or using yE/yF ratios). Table 2 (and later tables) – I did not find these tables very helpful, and wonder if it would be better to put this information onto the relevant figures as an additional plot (Lines 414-422 are very wordy as a result of describing Table 3 – would be easier to describe a graph of the same results). The figures on the table definitely do not need to be 2 decimal places.

We have kept the tables but changed the numbers to one digit after the decimal point.

10. Line 344: 'intentionally unrealistic' is ok, but readers might assume linear behavior between your realistic and unrealistic boundary conditions. Do you know if this is a reasonable assumption to make?

The parameter in question is fall velocity, which is not a boundary condition. While not linear, adaptation length monotonically increases with decreasing fall velocity.

11. Lines 451-2: I think (from memory – don't have the paper to hand) that this is similar to Philipps and Sutherland's formulation. If so, maybe reference this here.

The formulation of Phillips and Sutherland (1989) deals with bedload, not suspended load.

12. Lines 467-8: another way to interpret this is in terms of settling velocity (ie there is a settling velocity, or Stokes number, above which adaptation length can be ignored).

We agree with the reviewer that the adaptation can also be analyzed via settling velocity. But since settling velocity is a function of grain size, we think that it does not make much difference whether using settling velocity or grain size as the independent variable. Actually equation (31) (which is also exhibited in Figure 9) shows how adaptation length is controlled by the settling velocity.

13. Lines 555-7: I am not sure that the results in this paper can be used to make recommendations about event-scale modelling. It would be good to see some event-scale simulations to assess the significance of any differences between the two formulations.

Additional numerical cases with hydrographs have been added in the Supplement. Results indicate that our conclusions based on constant flow discharge also hold when hydrographs are implemented. See Sections S2 and S3 of the Supplement.

We have also moved Appendix A and Appendix B to the Supplement.

14. Line 28: I would say 'adapted for' rather than 'designed for'.

The manuscript has been revised as suggested by the reviewer.

15. Line 36: 'some aspect of sediment transport' is rather imprecise – can this be made more specific.

The manuscript has been rewritten to make the description more specific. The "aspect" is sediment transport itself. See Line 39 of the manuscript with track changes.

16. Line 119: presumably the grain-size distribution of the flux is also fixed?

Yes. Sediment of each grain size range is fed at a specific rate, which means the grain size distribution of the sediment supply is also fixed. The manuscript has been revised to make this issue clear. See Line 133 of the manuscript with track changes.

17. Lines 271-2: can the sorting of the gsd also be given (maybe use the blank spaces on Figure 2 to put these numbers onto)?

We have added the geometric standard deviation. See Lines 300-301 of the manuscript with track changes.

18. Line 281: I think it is 400 cells, but 401 computational nodes.

Since the finite volume method is implemented for the hydraulic calculation, our numerical method is based on cells rather than nodes. The channel length is 200 km, and the cell size is 0.5 km. Actually we have 401 cells with cell centers located at $x$ = 0 km, 0.5 km, 1 km, … , 199 km, 199.5 km, and 200 km.

19. Line 282: can you give Courant numbers for these timesteps?

The Courant number is defined as $(u+(g*h)^{0.5})*dth/dx$, where $u$ is flow velocity, $h$ is flow depth, and $dth$ is the time step for hydraulic calculation. The Courant number in all cases of our simulations is no more than 0.1.

20. Table 1: Add dx and dt to this – the table is a very useful quick reference point, so having these values here would be informative.

The values of $dx$ and $dt$ have been added to Table 1.

21. Figure 3: Could the water surface be added to this plot (maybe initial and final values only)?

Initial and final water surfaces have been added to Figures 3-8.

[revised manuscript text omitted]

---

## Referee Report (RR1)

Review by Astrid Blom (Oct 2018) of the _revised manuscript_

"Morphodynamic model of Lower Yellow River: flux or entrainment form for sediment mass conservation?"

by Chenge An et al.

The authors asses the differences between modelling sediment conservation using a flux-based formulation and an entrainment-deposition type formulation. They apply the two forms of the sediment conservation equation to the Lower Yellow River and study the differences between the predicted results. The work is novel and of interest to the ESurf reader. The differences between figures 6 and 7 are quite large, which is an interesting finding. I am quite happy with how the authors have addressed my previous concerns. I have two remaining suggestions: (1) to stress a bit more strongly the _essential_ difference between the flux form and the entrainment form of the conservation equation "_We define the flux form as based on the local capacity sediment transport rate, and the entrainment form as based on the local capacity entrainment rate_" and (2) explain the entrainment form of the conservation equation in more detail. I would suggest acceptance of the paper based on minor revision.

**Suggestions:**

_(Please note: line numbers and equation numbers all refer to numbers in the tracked changes manuscript)_

1.  Based on the authors' response to my previous 2[nd] comment I now understand what the authors mean by a flux based approach and an entrainment based approach. The authors' response is:

    > We have now specifically defined what we mean by the flux form and entrainment form. See Lines 23-24 of the manuscript with track changes. We define again here: "Here we identify the flux form as based on the local capacity sediment transport rate, and the entrainment form as based on the local capacity entrainment rate". Although we do not go into details in the paper, we note here that our system (the Yellow River) is suspension-dominated to the point where bedload is negligible. The phenomenon documented by Bell and Sutherland (1983) can be quantified using an entrainment-based _bedload_ formulation such as Pelosi and Parker (2014). We note here that El kadi Abderrezzak and Pacquier (2009) list the form $dQ_s/dx = (Q_s^{cap} - Q_s)/L_{ad}$, where $L_{ad}$ is the adaptation length. This is nothing more and nothing less than an entrainment form, but one for which the guts of the adaptation length $L_{ad}$ remain undefined. In our system $L_{ad}$ is specifically given as $q_w/(v_s r_0)$, where $q_w$ is the water discharge per unit width and $v_s$ is fall velocity.

    This response in the response letter helped me a lot, and I actually find it clearer than the explanation in the revised manuscript. I'd suggest incorporating the above lines in the manuscript text, also the part of Bell&Sutherland, Pelosi&Parker and El Kadi Abderrezzak.

    In trying to understand the essential difference between the flux form and the entrainment form, it was the following sentence by the authors that made the difference to me: "_Here we define the flux form as based on the local capacity sediment transport rate, and the entrainment form as_

*based on the local capacity entrainment rate.*" I'd stress that information at various points of the manuscript: abstract, conclusions and introduction. I indeed do find that sentence in the revised abstract, but I'd move it upward as it is quite an important remark that will help the reader understand the essential difference between the two forms.

2. Ln 54-55. The fact that one of the two (sediment transport rate or entrainment rate) is related to flow hydraulics is not the essence of the difference between the two forms, right? The essence rather is your phrase "*Here we define the flux form as based on the local capacity sediment transport rate, and the entrainment form as based on the local capacity entrainment rate.*"

3. Ln 25-26, 44. I find the terms equilibrium and nonequilibrium still confusing. I now understand that your words "(non)equilibrium" relate to the "sediment transport rate". Yet if you'd decide to relate the words "(non)equilibrium" to the "entrainment rate", would you have to reverse the use of the 2 words?

4. Also see my previous Comment 26. I'd add the information on the numerical schemes for solving the equation of conservation of sediment mass (in the section on the flux form as well as in the section describing the entrainment form) to the manuscript. I may have overlooked though.

5. Ln 152-153. "*For all the governing equations in this paper, the flood time scale is implemented by introducing If into each time derivative.*" I am not sure I understand the meaning of this information. I think it would be relevant to let the reader know explicitly which time coordinate, $t$ or $t_f$, is indicated in the legends of your figures? For instance, see figure 8. Maybe also list this information in the captions?

6. Section 2.3. I think the reader will need some help in this section.
   - Why not start off with listing the general entrainment form of the conservation equation of sediment mass, where the right-hand term of Eq.8 is equal to $D_s - E_s$ ?
   - then step by step explain what $D_s$ is and how it is modelled?
   - then $E_s$ ?
   - the information on slide 8 of Chapter 4 of Gary's E-book would definitely help here
   - I'd add references to the section
   - please also indicate it if an equation is newly introduced (and so no references exist)

7. Eq (9). I do not understand the background of this equation. Can you help me and the reader out here?

8. Ln 211. Please explain to the reader what is the physical meaning of the "sediment transport" being at equilibrium. I guess that under these conditions the upward sediment flux due to turbulent eddies is equal to the downward sediment flux related to gravity or the fall velocity. So under conditions in which the "sediment transport" is at equilibrium we are dealing with the Rouse profile?

9. Ln 674. Please note that the instabilities reported by Chavarrias et al (2018) do not have a numerical origin. They result from complex eigenvalues of the system of equations and do not result from the numerical solution procedure.

**Specific or detailed comments:**

10. Ln 49. alternate → alternative?

11. Ln 121. Reference to E&H consists of an unwanted ".", I think. Something like this is also found in L65.

12. Ln 183. Not sure you like how the symbol turns out here.

13. Ln 275. This text could use some more nuance.
    Indeed Blom et al (2016, GRL) apply the fractional form of E&H that was introduced by Van der Scheer et al (2002).
    Yet Blom et al (2017, JGR) propose a much more general power law load relation for mixed-size sediment (called "GR", but inspired on the form of E&H) that is capable of including hiding effects. It is listed in Eq 19 in Blom et al (2017 JGR).
    I think that the reason for it to not be applicable to the LYR is that it has not been calibrated to the LYR data and we do not know suitable values of the constants in the GR load relation. I do not think the reason for it to not be applicable to the LYR is that it does not include hiding effects.

14. Ln 281 and 305. Porosity value is listed twice.

**References**

Bell, R.G., Sutherland, A.J., 1983. Nonequilibrium bedload transport by steady flows. J. Hydraul. Eng. 109 (3). https://doi.org/10.1061/(ASCE)0733-9429(1983)109:3(351).

Blom, A., Viparelli, E., Chavarrías, V., 2016. The graded alluvial river: profile concavity and downstream fining. Geophys. Res. Lett. 43, 1–9. https://doi.org/10.1002/2016GL068898.

Blom, A., Arkesteijn, L., Chavarrías, V., Viparelli, E., 2017. The equilibrium alluvial river under variable flow and its channel-forming discharge. J. Geophys. Res. Earth Surf., https://doi.org/10.1002/2017JF004213.

Chavarrías, V., Stecca, G., and A. Blom, 2018. Ill-posedness in modeling mixed sediment river morphodynamics. Adv. Water Resour., doi:10.1016/j.advwatres.2018.02.011

Van der Scheer, P., J.S. Ribberink, and A. Blom (2002), Transport formulas for graded sediment; Behaviour of transport formulas and verification with data. Research Report 2002R-002, Civil Engineering, University of Twente, Netherlands

---

## Author Response (AR2)

**Responses to comments**

**Reviewer #1 (Astrid Blom)**

1. Based on the authors' response to my previous 2nd comment I now understand what the authors mean by a flux based approach and an entrainment based approach. The authors' response is:

"We have now specifically defined what we mean by the flux form and entrainment form. See Lines 23-24 of the manuscript with track changes. We define again here: "Here we identify the flux form as based on the local capacity sediment transport rate, and the entrainment form as based on the local capacity entrainment rate". Although we do not go into details in the paper, we note here that our system (the Yellow River) is suspension-dominated to the point where bedload is negligible. The phenomenon documented by Bell and Sutherland (1983) can be quantified using an entrainment-based bedload formulation such as Pelosi and Parker (2014). We note here that El kadi Abderrezzak and Pacquier (2009) list the form dQs/dx = (Qscap − Qs)/Lad. where Lad is the adaptation length. This is nothing more and nothing less than an entrainment form, but one for which the guts of the adaptation length Lad remain undefined. In our system Lad is specifically given as qw/(vsr0), where qw is the water discharge per unit width and vs is fall velocity."

This response in the response letter helped me a lot, and I actually find it clearer than the explanation in the revised manuscript. I'd suggest incorporating the above lines in the manuscript text, also the part of Bell&Sutherland, Pelosi&Parker and El Kadi Abderrezzak.

In trying to understand the essential difference between the flux form and the entrainment form, it was the following sentence by the authors that made the difference to me: "Here we define the flux form as based on the local capacity sediment transport rate, and the entrainment form as based on the local capacity entrainment rate." I'd stress that information at various points of the manuscript: abstract, conclusions and introduction. I indeed do find that sentence in the revised abstract, but I'd move it upward as it is quite an important remark that will help the reader understand the essential difference between the two forms.

We have moved the sentence upward in the abstract, as suggested by the reviewer. See Lines 20-21 of the manuscript with track changes. We have also stressed this sentence in the Conclusions and the Introduction. See Lines 57-59 and Lines 677-679 of the manuscript with track changes. We think that the response in the last response letter, as mentioned by the reviewer, actually has already been discussed in the third paragraph of the Introduction.

2. Ln 54-55. The fact that one of the two (sediment transport rate or entrainment rate) is related to flow hydraulics is not the essence of the difference between the two forms, right? The essence rather is your phrase "Here we define the flux form as based on the local capacity sediment transport rate, and the entrainment form as based on the local capacity entrainment rate."

The text has been revised as suggested by the reviewer. See Lines 57-59 of the manuscript with track changes.

3. Ln 25-26, 44. I find the terms equilibrium and nonequilibrium still confusing. I now understand that your words "(non)equilibrium" relate to the "sediment transport rate". Yet if you'd decide to relate the words "(non)equilibrium" to the "entrainment rate", would you have to reverse the use of the 2 words?

We think that we have been clear and consistent: the flux form corresponds to the equilibrium form, and the entrainment form corresponds to the nonequilibrium form.

4. Also see my previous Comment 26. I'd add the information on the numerical schemes for solving the equation of conservation of sediment mass (in the section on the flux form as well as in the section describing the entrainment form) to the manuscript. I may have overlooked though.

Information on the numerical schemes for solving the equation of conservation of sediment mass has been added for both the flux form and the entrainment form of the Exner equation. See Lines 188-189 and Lines 235-237 of the manuscript with track changes.

5. Ln 152-153. "For all the governing equations in this paper, the flood time scale is implemented by introducing If into each time derivative." I am not sure I understand the meaning of this information. I think it would be relevant to let the reader know explicitly which time coordinate, t or tf, is indicated in the legends of your figures? For instance, see figure 8. Maybe also list this information in the captions?

We have rewritten the sentence to make this issue clearer. We also now state explicitly that "results we exhibit later in this paper are all cast in terms of actual time scale t". See Lines 148-152 of the manuscript with track changes.

6. Section 2.3. I think the reader will need some help in this section.
- Why not start off with listing the general entrainment form of the conservation equation of sediment mass, where the right-hand term of Eq.8 is equal to Ds - Es ?
- then step by step explain what Ds is and how it is modelled?
- then Es ?
- the information on slide 8 of Chapter 4 of Gary's E-book would definitely help here
- I'd add references to the section
- please also indicate it if an equation is newly introduced (and so no references exist)

We did not add a new equation here, but rather we added a sentence to explain the two terms on the right hand side of Eq. (8). One of the terms denotes the dimensional entrainment rate and the other denotes the dimensional deposition rate. We think that this explanation should be enough for the reader to understand. See Lines 197-199 of the manuscript with track changes.

7. Eq (9). I do not understand the background of this equation. Can you help me and the reader out here?

Information has been added here to explain the equation for dimensionless entrainment rate E.

The basic idea is that "sediment transport reaches its equilibrium state ($q_s = q_{se}$) when the sediment deposition rate and the sediment entrainment rate balance each other ($r_0 C = E$)". See Lines 205-214 of the manuscript with track changes.

8. Ln 211. Please explain to the reader what is the physical meaning of the "sediment transport" being at equilibrium. I guess that under these conditions the upward sediment flux due to turbulent eddies is equal to the downward sediment flux related to gravity or the fall velocity. So under conditions in which the "sediment transport" is at equilibrium we are dealing with the Rouse profile?

We have explained in the manuscript that the condition of sediment transport being at equilibrium corresponds to "$q_s = q_{se}$"(i.e. the sediment transport rate equals the sediment transport capacity). See Line 209 of the manuscript with track changes.

9. Ln 674. Please note that the instabilities reported by Chavarrias et al (2018) do not have a numerical origin. They result from complex eigenvalues of the system of equations and do not result from the numerical solution procedure.

We have revised the manuscript to render it more accurate. See Lines 673-674 of the manuscript with track changes.

10. Ln 49. alternate → alternative?

The manuscript has been revised as suggested by the reviewer.

11. Ln 121. Reference to E&H consists of an unwanted ".", I think. Something like this is also found in L65.

We appreciate the reviewer for the correction. The manuscript has been revised.

12. Ln 183. Not sure you like how the symbol turns out here.

We have rewritted the symbol as $f_i\big|_{z_b - L_a}$ so as to show the two layers of subscripts.

13. Ln 275. This text could use some more nuance. Indeed Blom et al (2016, GRL) apply the fractional form of E&H that was introduced by Van der Scheer et al (2002). Yet Blom et al (2017, JGR) propose a much more general power law load relation for mixed-size sediment (called "GR", but inspired on the form of E&H) that is capable of including hiding effects. It is listed in Eq 19 in Blom et al (2017 JGR). I think that the reason for it to not be applicable to the LYR is that it has not been calibrated to the LYR data and we do not know suitable values of the constants in the GR load relation. I do not think the reason for it to not be applicable to the LYR is that it does not include hiding effects.

We have revised the text to make it more accurate. We now state that the Engelund-Hansen equation for mixtures as implemented by Van der Scheer et al. (2002) and Blom et al. (2016, 2017) is not applicable to the LYR because it has not been calibrated to the LYR data. See Lines 282-284 of the manuscript with track changes.

14. Ln 281 and 305. Porosity value is listed twice.

The sentence which lists the porosity value for the second time has been deleted. See Line 313 of the manuscript with track changes.

**Reviewer #3**

1. My primary suggestion concerns the presentation of the results comparing the two forms of the Exner equation – i.e., the "delta" metric. I think it is fine and appropriate to calculate the relative difference of the different variables predicted by the two models, but I have a concern that as framed, the delta metric may over- or (especially) under-represent the relative difference between the two results. For example, delta(zb) is shown to be within about 4% after 0.2 years for the uniform sediment simulations. This will be highly sensitive to the choice of elevation datum used in the model; i.e., if the initial upstream bed elevation is 1000 m instead of 20 m, then the 2.3 vs. 3 m of scour observed would lead to a delta(zb) of about 0.07%! I recommend calculating the delta values with the deviation from the initial condition, rather than the final value, for both the elevation and other variables (grain size, sediment transport rate). If that were done, then the delta(zb) for the uniform case would be (3-2.3)/3 = 23%, instead of 4%. Calculating results this way may change some of the story, but I suspect the overall message would remain the same.

We admit that the values of delta(zb) depend on the choice of elevation datum. However, such a problem does not exist for the values of delta(qs), delta(Dsg) and delta(Dlg). The referee suggests that we calculate the delta values with the deviatoric bed elevation (dzb) from the initial condition rather than bed elevation (zb) itself. This leads to a new problem. If the flux form predicts a deviatoric elevation of 0 m (or a very small value as in the numerical model), whilst the entrainment form predict a deviatoric elevation of 0.01 m, then delta(dzb) will be infinitely large. But the difference between the predictions of the two forms is actually quite small (0 m vs. 0.01 m). For this reason, we have not replaced delta(zb) with delta(dzb) as suggested by the reviewer.

What we have done in the manuscript is as follows: (1) we state that the value of delta(zb) depends on the choice of elevation datum, and that the downstream bed elevation is fixed as 0 m in this paper; and (2) we state that the maximum values of delta(zb) are almost always realized at the upstream end, where the bed elevation nevertheless does not deviate far from the initial value of 20 m. See Lines 377-380 of the manuscript with track changes. We hope this helps the reader understand the difference in bed elevation predicted by the two forms of the Exner equation.

2. A major motivating question for this paper is which version of the Exner equation is appropriate. The authors provide some guidance, suggesting that the adaptation length scale of the sediment in the system should be an important consideration, and I think their reasoning is sound. Some additional comparison with actual data from the Lower Yellow River would be helpful as well though – Does the LYR show evidence of sorting waves, etc., which could provide additional

We have consulted our coauthor, Yuanfeng Zhang, who is perhaps the foremost (in China and thus the world) expert on sediment transport in the Yellow River. He says that the adjustment of grain size distribution of the LYR is quite smooth in space and time. No abrupt change of grain size distribution has been observed in the LYR. This serves as an indirect evidence that there are no clear sorting waves in the LYR. We have stated this in the manuscript. See Lines 629-631 of the manuscript with track changes.

3. Line 20: "local function of bed shear stress" = although it is common to use bed shear stress to calculate sediment transport rates, it is not the only possibility (i.e., one could use velocity, stream power, etc.). Suggest being a bit more general, perhaps "sediment transport is a function of local hydraulic conditions" or something.

The manuscript has been revised as suggested by the reviewer. See Lines 21-22 of the manuscript with track changes.

4. Line 21: "equilibrium" – suggest clarifying that here you mean equilibrium with hydraulic conditions (as opposed to local sediment balance).

The manuscript has been revised as suggested by the reviewer.

5. Line 22: "identify" is a strange word here – maybe use "represent"?

The manuscript has been revised as suggested by the reviewer.

6. Line 27: should be noted in the abstract that you are using a one-dimensional morphodynamic model.

Information has been added to the abstract. See Line 31 of the manuscript with track changes.

7. Line 33: Related to the general comment above – it would benefit the abstract to have a statement about which version of the Exner equation is appropriate in what circumstances.

Information has been added to the abstract. See Lines 37-38 of the manuscript with track changes.

8. Line 43: see comment above about local bed shear stress (line 20).

The manuscript has been revised as suggested by the reviewer. See Lines 47-48 of the manuscript with track changes.

9. Line 50: "pioneering work *on* bedload transport"

The manuscript has been revised accordingly.

10. Line 166: "summer" should be "summed"

The manuscript has been revised accordingly. We appreciate the reviewer for the correction.

11. Line 195: "form" should be "from".

The text has been rewritten accordingly.

12. Line 260: double period at the end of the sentence

The manuscript has been revised accordingly.

13. Line 263: Can you give some justification as to why a relation without a hiding function is not appropriate for the LYR?

We have revised the text as suggested by Reviewer #1. We now state that the Engelund-Hansen equation for mixtures as implemented by Van der Scheer et al. (2002), Blom et al. (2016, 2017) is not applicable to the LYR because it has not been calibrated to the LYR data. See Lines 282-284 of the manuscript with track changes.

14. Line 292: The porosity used in the modeling was already given in line 268.

We have deleted the sentence. Thanks for the correction.

15. Lines 377-379: I am not convinced you need to include these sentences – just state that the supplement looks at hydrographs.

The sentences have been deleted as suggested by the reviewer.

16. Line 119: presumably the grain-size distribution of the flux is also fixed?

As we state in Lines 415-419 of the manuscript with track changes, the grain size distribution of the sediment feed (i.e. flux at upstream end) is fixed. It is not fixed in other locations, but is free to adjust morphodynamically.

17. Line 390: 173.7 Mt/a – is this correct? Adding washload nearly doubles the sediment load? Also, this value is larger than the stated range of 89-126 Mt/a.

Naito et al. (accepted subject to revision) estimate that 45% of the suspended load of LYR is washload based on data at the Huayuankou gauging station. This number agrees well with Fig. 1c of our paper, in which washload (finer than 45 μm) make up about 40% of the suspended load for various gauging stations of the LYR.

18. Figures 6, 7, and 8: The vertical axes on the grain size graphs say the values are in mm, not um.

The vertical axes of Figures 6, 7 and 8 have been corrected. The vertical axes of Figure S5 and S6 in the supplementary information have also been corrected. Thank you!

[revised manuscript text omitted]
} (1 - \lambda_p) \left[ f_{Ii} \frac{\partial}{\partial t} (z_b - L_a) + \frac{\partial}{\partial t} (F_i L_a) \right] = -v_{si} (E_i - r_{0i} C_i) \hspace{3cm} (12)$$

$$E_i = r_{0i} \frac{q_{sei}}{q_w} \hspace{11cm} (13)$$

where the subscript $i$ denotes the $i$-th size range of sediment grain size.

Summing Eq. (12) over all grain size ranges, we get the governing equation for bed elevation,

$$\frac{1}{I_f}\left(1-\lambda_p\right)\frac{\partial z_b}{\partial t} = -\sum_{j=1}^{n} v_{sj}\left(E_j - r_{0j}C_j\right)$$
(14)

Reducing Eq. (12) with Eq. (14) we get the governing equation for surface fraction $F_i$,

$$\frac{1}{I_f}\left(1-\lambda_p\right)\left[L_a\frac{\partial F_i}{\partial t} + \left(F_i - f_{Ii}\right)\frac{\partial L_a}{\partial t}\right] = f_{Ii}\sum_{j=1}^{n} v_{sj}\left(E_j - r_{0j}C_j\right) - v_{si}\left(E_i - r_{0i}C_i\right)$$
(15)

The governing equation for the sediment concentration of each grain size $C_i$ can be written as,

$$\frac{1}{I_f}\frac{\partial\left(hC_i\right)}{\partial t} + \frac{\partial\left(huC_i\right)}{\partial x} = v_{si}\left(E_i - r_0C_i\right)$$
(16)

and the sediment transport rate per unit width for the $i$-th size range $q_{si}$ obeys the following continuity relation,

$$q_{si} = huC_i$$
(17)

In the entrainment formulation, the closure relation for $f_{Ii}$ is the same as that used in the flux formulation (i.e., Eq.
(7)), and the substrate stratigraphy is also stored and accessed using the method of Viparelli et al. (2010). When discretizing
the entrainment form of the Exner equation, a first-order upwinded scheme is implemented for the spatial derivatives, and a
first-order explicit scheme is implemented for the temporal derivatives.

**2.4 Sediment transport relation**

**2.4.1 Uniform sediment**

To close the Exner equations described in Sections 2.2 and 2.3, equations for equilibrium sediment transport rate $q_{se}$
($q_{sei}$) are still needed. For the simulations using uniform sediment, we implement the generalized Engelund-Hansen relation
proposed by Ma et al. (2017). This equation is based on the data from LYR and can be written in the following dimensionless
form,

$$q_s^* = \frac{\alpha_s}{C_f}\left(\tau^*\right)^{n_s}$$
(18)

where $q_s^*$ is dimensionless sediment transport rate per unit width (i.e., the Einstein number), and $\tau^*$ is dimensionless shear
stress (i.e., the Shields number). They are defined as,

$\qquad q_s{}^* = \dfrac{q_{se}}{\sqrt{RgD}D}$ (19)

$\qquad \tau^* = \dfrac{\tau_b}{\rho RgD}$ (20)

$\qquad \tau_b = \rho C_f u^2$ (21)

where $D$ is the characteristic grain size of the bed sediment (here approximated as uniform); $\tau_b$ is bed shear stress; and $R$ is submerged specific gravity of sediment, defined as $(\rho_s - \rho) / \rho$, in which $\rho_s$ is density of sediment, and $\rho$ is density of water.

The sediment submerged specific gravity $R$ is specified as 1.65 in this paper, which is an appropriate estimate for natural rivers, and corresponds to quartz.

$\qquad$ In the relation of Ma et al. (2017), the dimensionless coefficient $\alpha_s = 0.9$ and the dimensionless exponent $n_s = 1.68$.

These values are quite different from the original relation of Engelund and Hansen (1967), in which $\alpha_s = 0.05$ and $n_s = 2.5$.

Ma et al. (2017) demonstrated that such differences imply that the riverbed of the LYR is dominated by low-amplitude bedform features (dunes) approaching upper-regime plane bed. According to this finding, form drag is then neglected in our modeling, and all of the bed shear stress is used for sediment transport.

**2.4.2 Sediment mixtures**

$\qquad$ We implement the relation of Naito et al. (accepted subject to revision) to calculate the equilibrium sediment transport rate of size mixtures. Using field data from the LYR, Naito et al. (accepted subject to revision) extended the Engelund and

Hansen (1967) relation to a surface-based grain-size specific form, in which the suspended load transport rate of the $i$-th size range is tied to the availability of this size range on the bed surface:

$\qquad q_{sei} = \dfrac{N_i^* F_i u_*^3}{RgC_f}$ (22)

where $N_i^*$ is the dimensionless sediment transport rate in the $i$-th size range, and $u_*$ is shear velocity calculated from the bed shear stress $\tau_b$:

$\qquad u_* = \sqrt{\dfrac{\tau_b}{\rho}}$ (23)

$\qquad$ The transport relation itself takes the form,

[revised manuscript text omitted]